# GRAPH CLUSTERING WITH THE WEAK SUPERVISION FROM GRAPH LABELS

## ABSTRACT

Graph Clustering, which clusters the nodes of a graph given its collection of node features and edge connections in an unsupervised manner, has long been researched in graph learning and is essential in certain applications. While this task is common, more complex cases arise in practice—can we cluster nodes better with some graph-level side information or in a weakly supervised manner as, for example, identifying potential fraud users in a social network given additional labels of fraud communities. This triggers an interesting problem which we define as *Weakly Supervised Graph Clustering* (WSGC). In this paper, we firstly discuss the various possible settings of WSGC, formally. Upon such discussion, we investigate a particular task of weakly supervised graph clustering by making use of the graph labels and node features, with the assistance of a hierarchical graph that further characterizes the connections between different graphs. To address this task, we propose Gaussian Mixture Graph Convolutional Network (GMGCN), a simple yet effective framework for learning node representations under the supervision of graph labels guided by a proposed consensus loss and then inferring the category of each node via a Gaussian Mixture Layer (GML). Extensive experiments are conducted to test the rationality of the formulation of weakly supervised graph clustering. The experimental results show that, with the assistance of graph labels, the weakly supervised graph clustering method has a great improvement over the traditional graph clustering method.

## 1 INTRODUCTION

With the development of deep graph learning, graph clustering has achieved significant improvement in community detection (Jin et al., 2019), text clustering (Aggarwal & Zhai, 2012) and other applications (Yang et al., 2010). Such applications all assume that only node/edge information and connections are accessible to assist with graph clustering in an unsupervised manner. However, it is possible that more graph-level side information, such as graph labels, may also be accessible and helpful for clustering the nodes in graphs. Or namely, *can we use the global label of the graph to help improve the effectiveness of graph clustering?*

This problem is interesting and very common in the real world. For example, point cloud segmentation, which could be re-considered as clustering of nodes in point cloud if we only know the objects contained in each point cloud but not all the node labels in the training process instead of predictions for nodes with known node labels that are solved by supervised graph learning methods (Ye et al., 2018; Li et al., 2019). Moreover, in social media, thinking of hot events that are widely discussed in the form of graphs where users are involved as nodes and the mutual following relationship among users as edges, it is interesting to think how to pick out the malicious users from the mass of users with the labels of these event topics such as the authenticity of each topic (Zhou & Zafarani, 2018; Cao et al., 2018). Actually, this problem is more like graph clustering in a weakly supervised manner (Zhou, 2018), or named *Weakly Supervised Graph Clustering* (WSGC), where the available training labels are much coarser-grained than the node labels we want to fit. Unlike traditional graph clustering (Girvan & Newman, 2002) which only conducts clustering based on node features or graph structures, WSGC is better guaranteed by the given label information of graphs, which, we will demonstrate, closely influences the clustering results.

Note that WSGC is different from traditional weakly supervised learning problems. Generally, weakly supervised learning is considered to consist of three types of weak supervision problems: incomplete supervision, inaccurate supervision and inexact supervision (Zhou, 2018). WSGC is different from incomplete supervision where a subset of training nodes is given with labels and inaccurate supervision where the given node labels are not always ground-truth, since WSGC does not contain any node labels for training. It is also different from inexact supervision, whose representative method as Multiple-Instance Learning (MIL) (Carbonneau et al., 2018) adopts global labels of bags to identify the labels of local instances. Because MIL assumes that each instance is i.i.d. and there are no edge connections involved. As far as we know, there are no studies on WSGC even though WSGC is very useful to formulate some real-world problems. Therefore, it is interesting and important to study the WSGC as a new challenge and seek new solutions for it.

In this paper, we formally define *Weakly Supervised Graph Clustering* (WSGC), i.e. a graph clustering task by making use of graph labels, graph structure and node features. To address this particular task, we propose a novel model based on Gaussian mixture model (GMM) and graph convolutional network (GCN) named as Gaussian Mixture Graph Convolutional Network (GMGCN). The proposed model includes a Gaussian mixture layer (GML) and a new hierarchical GAT (hierGAT) to update the node and graph hidden representations, respectively. Specifically, we design a consensus loss that plays a key role in improving node clustering with the assistant of graph labels in the training process. Finally, a graph clustering is carried out according to the parameters of GML. The main contributions are as follows:

1. **New problem:** We introduce a new problem, i.e. *Weakly Supervised Graph Clustering* (WSGC), which tries to identify the nodes in graphs based on the labels of graphs, and we take a formal discussion of the variants of this problem to attract more research attention on this problem. 2. **New solution:** We propose an effective model called GMGCN based on GMM and GCN to solve WSGC. To the best of our knowledge, this is the first work to achieve graph clustering in graph structure by integrating GMM into GCN. 3. **New loss:** We propose a consensus loss function to boost the model training through the principle of "same attraction, opposite repulsion". Experiments validate the effectiveness of this consensus loss.

We validate the significant improvement of WSGC over traditional clustering methods on various synthetic and real-world datasets. Compared to the best performing baseline method, GMGCN improves Normalized Mutual Information[1] (NMI) by an average of 13% on 5 synthetic datasets, by more than 13% on PHEME social media datasets, and improves mean intersection-over-union[2] (mIoU) by more than 18% on point cloud segmentation datasets.

## 2   Related Works

**Graph Learning.** Recently, there has been an increasing interest in the graph learning domain. Among all the existing works, GCN is one of the most effective convolution models. A typical GCN model is the message passing neural network (MPNN) proposed by Gilmer *et al.* (Gilmer et al., 2017) which re-generalizes several neural network and graph convolutional network approaches as a general "message-passing" architecture. Many kinds of GCN (Bruna et al., 2014; Defferrard et al., 2016; Kipf & Welling, 2017; Chang et al., 2020) deliver different message propagation functions for GCN. Among them, Graph Attention Networks (GAT) (Veličković et al., 2017) first leverages learnable self-attentional layers to aggregate weighted neighbors' information. Besides these methods to obtain the appropriate node representation, pooling strategies are proposed to integrate information over the node representations (Wu et al., 2020), such as max/min pooling (Defferrard et al., 2016), SortPooling (Zhang et al., 2018), and so on. In addition, Lin *et al.* (Lin et al., 2017) propose a learnable attentive pooling for weighted averaging of node representations. Despite extensive research on node-level and graph-level graph learning tasks, few studies have been conducted on WSGC.

**Graph Clustering.** Graph clustering is a fundamental data analysis task that aims to group similar nodes into the same category. Many real-world applications are cast as graph clustering (Shi & Ma-

---

[1]NMI is a normalization of the Mutual Information score to scale the clustering results between 0 (no mutual information) and 1 (perfect correlation).

[2]IoU is a commonly used metric in semantic segmentation measures similarity between finite sample sets, and is defined as the size of the intersection divided by the size of the union of the sample sets.

lik, 2000; Hastings, 2006). The major strategy of graph clustering is to perform traditional clustering algorithms such as $K$-means (Jain, 2010) or GMM (McLachlan & Basford, 1988) on the node features. With the great achievements of deep learning, more graph clustering studies have resorted to graph learning to learn embedding that captures both node features and structural relationships (Wu et al., 2020). Researchers employ the stacked sparse autoencoder (Tian et al., 2014), the variational autoencoder (Kipf & Welling, 2016), or the combination of both autoencoder and GCN (Bo et al., 2020) to obtain graph representations for clustering. Nevertheless, these graph clustering methods do not perform better than WSGC methods due to the lack of attention to graph labels.

**Multiple-Instance Learning.** Another task that has a similar definition to WSGC is Multiple-Instance Learning (MIL). MIL is a variant of inductive machine learning, where each learning example consists of a bag of instances instead of a single feature vector (Foulds & Frank, 2010). When obtaining local instances annotations is costly or not possible, but global labels for bags are available, MIL is utilized to train classifiers using weakly labeled data. It has received a considerable amount of attention due to its applicability to real-world problems (Andrews et al., 2002a; Cheplygina et al., 2019). Although MIL learns a classifier from the labels of bags to classify instance, it ignore structural information between instances and are therefore not suitable for WSGC.

**Gaussian Mixture Model.** As a core part of the proposed GMGCN model, GMM is a parametric probability density function for a weighted sum of Gaussian component densities (McLachlan & Basford, 1988). It is commonly used to find underlying clusters in data samples (Bishop, 2006). Generally, the GMM parameters are estimated using the iterative Expectation-Maximization (EM) (Moon, 1996) algorithm from training data. In this paper, we integrate the GMM into GCN to establish a solution for WSGC and update the GMM parameters using stochastic gradient descent.

**Point Cloud Semantic Segmentation.** Semantic segmentation of 3D point clouds has attracted more researchers' attentions due to the developments of 3D scanners, such as Light Detection and Ranging (LIDAR), structure-from-motion (SFM) techniques, etc. To operate directly on these unstructured point clouds, PointNet (Qi et al., 2017a) is proposed to subdivide the input points into a grid of blocks and process each such block individually, but limited neighborhood context of each point is taken into consideration. To overcome this drawback, many approaches (Engelmann et al., 2017; Ye et al., 2018; Qi et al., 2017b) are built upon PointNet and better capture local spatial structures as well as long dependency context. Recently, some latest works (Qi et al., 2017c; Landrieu & Simonovsky, 2018; Wang et al., 2019) re-consider semantic segmentation of unstructured 3D point clouds as predictions for nodes on graph data and incorporate contextual information and topological information on top of 3D points. However, these supervised graph learning methods are not suitable for weakly supervised point cloud semantic segmentation datasets if the point labels are unknown.

## 3 NOTATIONS AND PROBLEM STATEMENT

### 3.1 NOTATIONS

We denote a set of graph instances as $G = \{(\mathcal{G}^{(1)}, y^{(1)}), \ldots, (\mathcal{G}^{(N)}, y^{(N)})\}$ with $N$ graphs, where $\mathcal{G}^{(n)}$ refers to the $n$-th graph instance and $y^{(n)} \in \{0, 1, \ldots, C_{graph} - 1\}$ is the corresponding graph label of $C_{graph}$ different categories. We denote by $\mathcal{G}^{(n)} = (\mathbb{V}^{(n)}, \mathcal{E}^{(n)})$ the $n$-th graph instance of size $M_n$ with nodes $v_i^{(n)} \in \mathbb{V}$ and edges $(v_i^{(n)}, v_j^{(n)}) \in \mathcal{E}^{(n)}$, by $\mathbf{X}^{(n)} = \{x_1^{(n)}, \ldots, x_{M_n}^{(n)}\} \in \mathbb{R}^{M_n \times d}$ the feature matrix of nodes $\mathbb{V}^{(n)}$ where $d$ denotes the number of dimensions of node original features, and by $\mathbf{A}^{(n)} \in \{0, 1\}^{M_n \times M_n}$ the adjacency matrix which associate edge $(v_i^{(n)}, v_j^{(n)})$ with $A_{i,j}^{(n)}$. We denote by $\{z_1^{(n)}, \ldots, z_{M_n}^{(n)}\} \in \{0, 1, \ldots, C_{node} - 1\}$ the potential labels of nodes $\mathbb{V}^{(n)}$ where $C_{node}$ is the expected number of node categories. In addition, we denote by $\mathbf{A}^{hier} \in \{0, 1\}^{N \times N}$ the adjacency matrix of the links among graphs if the graphs in $G$ contain interconnections.

### 3.2 PROBLEM STATEMENT

The *Weakly Supervised Graph Clustering* (WSGC) problem is defined as: Given a set of graph-label pairs $G$ and node features $\mathbf{X}$, how to infer the label of the $i$-th node in the $n$-th graph, $z_i^{(n)}$, where $n \in \{1, \ldots, N\}$ and $i \in \{1, \ldots, M_n\}$? Under the definition of WSGC, there are some different cases that could be extended from the problem as new tasks.

**Case 1 (Infer Original Nodes)** *Given the whole $G$ and $\mathbf{X}$ as training data, how to infer all the node labels $z_i^{(n)}$ in the $\mathcal{G}^{(n)}s$?*

**Case 2 (Infer New Nodes)** *Given the portion of nodes $\mathbb{V}_{train}^{(n)} = \{v_1^{(n)}, \ldots, v_{M_n-s}^{(n)}\}$ in each $\mathcal{G}^{(n)}$ as training data, how to infer the labels of the rest of $s$ nodes $\{v_{M_n-s+1}^{(n)}, \ldots, v_{M_n}^{(n)}\}$ in the $\mathcal{G}^{(n)}s$?*

**Case 3 (Infer New Graphs)** *Given the portion of graphs $G_{train} = \{(\mathcal{G}^{(1)}, y^{(1)}), \ldots, (\mathcal{G}^{(N-S)}, y^{(N-S)})\}$ as training data, how to infer the node labels $z_i^{(n)}, n \in \{N - S + 1, \ldots, N\}$ in the rest of $S$ graphs, $\{\mathcal{G}^{(N-S+1)}, \ldots, \mathcal{G}^{(N)}\}$?*

**Case 4 (With/Without Hierarchical Graph)** *Given the interconnections among graphs as $\mathbf{A}_{hier}$, how to infer the node labels? And if there are no interconnections, how to infer the node labels (i.e. $\mathbf{A}_{hier} = \mathbf{I}$)?*

## 4 PROPOSED METHOD

In this section, we first introduce the preliminaries related to the proposed model, then we discuss details of the proposed method.

### 4.1 PRELIMINARIES

**Graph Attention Networks.** Graph Attention Networks (GAT) (Veličković et al., 2017) has been widely adopted in the field of graph convolution due to the learnable attentions to neighbors during the message passing process. A multi-head GAT Convolutional layer (GATConv) is formulated as:

$$\vec{h}_i = \overset{K}{\underset{k=1}{\|}} \sigma(\alpha_{i,i}^k \mathbf{\Theta} \vec{x}_i + \sum_{j \in N(i)} \alpha_{i,j}^k \mathbf{\Theta} \vec{x}_j), \tag{1}$$

where $\vec{h}_i$ and $\vec{x}_i$ refer to the hidden feature representation for the next layer and the current hidden feature vector of node $i$ in the graph, respectively. $\|$ refers to a concatenation operation for $K$-head attention, and $\sigma(\cdot)$ refers to the nonlinear activation function. $\mathbf{\Theta}$ is the weight matrix and $N(i)$ is the set of neighbors of node $i$. $\alpha_{i,j}^k$ is the attention coefficient computed by the $k$-th attention mechanism as follows:

$$\alpha_{i,j} = \frac{\exp(\mathrm{LeakyReLU}(\vec{\mathrm{a}}^\top [\mathbf{\Theta} \vec{x}_i \| \mathbf{\Theta} \vec{x}_j]))}{\sum_{l \in N(i) \cup \{i\}} \exp(\mathrm{LeakyReLU}(\vec{\mathrm{a}}^\top [\mathbf{\Theta} \vec{x}_i \| \mathbf{\Theta} \vec{x}_l]))}, \tag{2}$$

where attention mechanism is a single-layer feedforward neural network parametrized by a weight vector $\mathrm{a}$ and applying the LeakyReLU nonlinearity.

**Graph Attentive Pooling.** Lin *et al.* (Lin et al., 2017) employ a self-attentive mechanism to learn the importance of each node and then transform a variable number of node representations into a fixed-length graph representation:

$$h_g^{(n)} = \mathrm{Attn}(\mathbf{H}^{(n)}) = \mathrm{softmax}(\mathbf{\Theta}_{s2}\mathrm{tanh}(\mathbf{\Theta}_{s1}\mathbf{H}^{(n)\top}))\mathbf{H}^{(n)}, \tag{3}$$

where $\mathbf{\Theta}_{s1}$ and $\mathbf{\Theta}_{s2}$ refer to trainable parameters. The unified graph representation of the $n$-th graph $h_g^{(n)}$ is obtained by multiplying the node attention scores with $\mathbf{H}^{(n)}$.

### 4.2 OVERALL FRAMEWORK

Fig. 1 gives an overall framework of the proposed method to infer the categories of the nodes in the graphs. It consists of five processes: node-level modeling, node feature redistribution, graph instance modeling, hierarchical graph modeling and consensus loss calculation. The node-level modeling consists of 2-layer Multi-Head GAT and the graph instance modeling is accomplished by aggregating graph-level representations from the node-level GML via graph attentive pooling. In this section, we will introduce the rest of the three key components of the proposed framework.

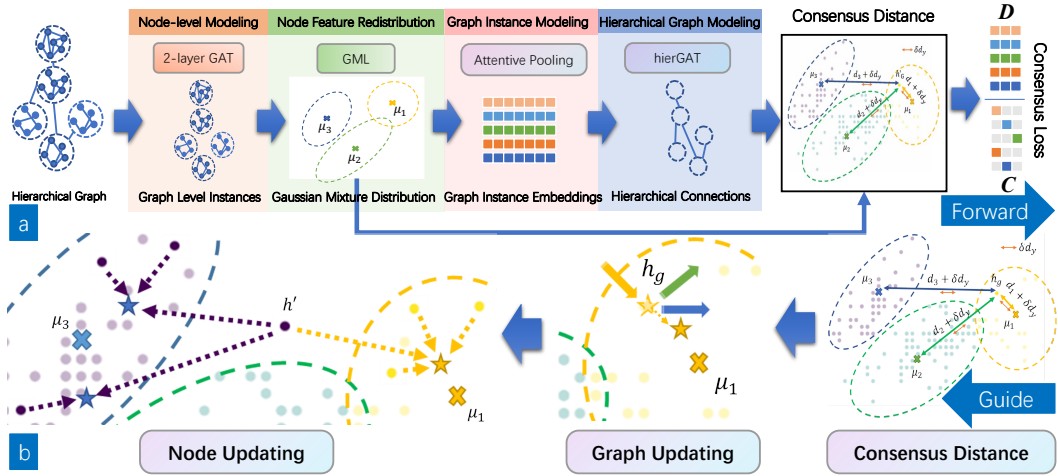

Figure 1: The overall framework of the proposed GMGCN. (a) Model forward propagation consists of five processes: node-level modeling, node feature redistribution, graph instance modeling, hierarchical graph modeling and consensus loss calculation. (b) During the backward propagation, the distances between graph representations $h_g$ obtained via hierGAT and the mean vector $\mu$ of each Gaussian weight function will guide graph updating and further influence the node updating.

**Gaussian Mixture Layer (GML).** As the key component of the GMGCN, GML plays a crucial role in the process of node feature redistribution. It is described as $\vec{h}'_i = \frac{1}{C_{graph}} \sum_{c=1}^{C_{graph}} w_c(\vec{h}_i)$, where $w_c(\cdot)$ represents the Gaussian weight function as follows:

$$w_c(\vec{h}_i) = \exp(-\frac{1}{2\vec{\sigma}_c^2}(\Theta_{GML}\vec{h}_i - \vec{\mu}_c)^2), \tag{4}$$

where $\vec{\sigma}_c$ and $\vec{\mu}_c$ are learnable standard deviation vector and mean vector, respectively, and $\Theta_{GML}$ refers to the learning parameters.

Based on the idea of GMM, GML has the ability to effectively distinguish different types of nodes according to the input features. However, it could not resolve two major challenges of WSGC: the node-level modeling and the graph instance modeling that affects the final node inference. Therefore, we employ the node representations from the 2-layer GAT as inputs of GML. The node representations provide local structural information of the nodes to GML. Meanwhile, the outputs $\vec{h}'_i$ of GML are aggregated by attentive pooling as $h_g^{(n)}$, and further fed into a hierarchical GAT.

**Hierarchical GAT (hierGAT).** As mentioned in *Case 4*, the information of interconnections among graphs may infer the node labels. To model such hierarchical structure information, we employ a 3-layer hierGAT to update the feature representations in the hierarchical graph as Eq. (5).

$$\mathbf{H}_g^{l+1} = \text{hierGAT}(\mathbf{H}_g^l) = \beta\boldsymbol{\alpha}^{hier}\mathbf{H}_g^l + (1-\beta)\mathbf{H}_g^0, \tag{5}$$

where $l \in \{0, 1, 2\}$ refers to the layer number of the 3-layer hierGAT, $\boldsymbol{\alpha}^{hier}$ refers to the hierarchical attention coefficients, and $\mathbf{H}_g^0 = \mathbf{H}_g$, which is the hierarchical feature matrix that consists of the feature representations of graphs, $h_g^{(n)}$. Each hierarchical attention coefficient $\alpha_{ij}^{hier}$ in $\boldsymbol{\alpha}^{hier}$ is caluculated as Eq. (2). We define the final output of the 3-layer hierGAT $\mathbf{H}_g^3$ as $\mathbf{H}'_g$. There are no training parameters that causes the spatial transformation in Eq. (5) so that the node embeddings $\mathbf{H}'^{(n)}$ and graph embeddings $\mathbf{H}'_g$ are still in the same hidden space. Note that, the hierGAT is reduced to a version without using the hierarchical graph when $\beta = 0$.

**Consensus Loss.** The core idea of consensus loss is to make the graph representations with the same labels closer, and those with different labels farther away in the hidden space. Therefore, before computing the consensus loss, we first compute the similarity matrix $\mathbf{S} \in \mathbb{R}^{N \times C_{graph}}$ of the graph representations, which is obtained by the following steps:

First, we compute the distance between the graph representation $\vec{h}'^{(n)}_g$ obtained via hierGAT and the mean vector $\vec{\mu}_c$ of $c$-th Gaussian weight function as follows:

$$d_{nc} = \| \vec{h}'^{(n)}_g - \vec{\mu}_c \|_2 . \tag{6}$$

Second, we enhance the distance of the graph representations to the true mean vector on $d_{nc}$ by:

$$(d_{nc})_{enhance} = d_{nc} + \delta d_{ny^{(n)}}, \qquad (7)$$

where $\delta$ is a discount hyperparameter, $y^{(n)}$ is the true label of graph $\mathcal{G}^{(n)}$, and $d_{ny^{(n)}}$ represents the distance between the graph representation $\vec{h}'^{(n)}_g$ and the mean vector $\vec{\mu}_{y^{(n)}}$. Then the similarity matrix $\mathbf{S}$ is formulated as $\mathbf{S} = \text{softmax}(-\mathbf{D}_{enhance})$, where $\mathbf{D}_{enhance}$ consists of $(d_{nc})_{enhance}$. Finally, the consensus loss is formulated as $l_{con} := cross\_entropy(\mathbf{S}, \mathbf{y})$.

**Model training Guidance.** As shown in Fig. 1(b), the consensus loss makes the graph representation of each graph (e.g. $h_g$) closer to the mean vector of the Gaussian function corresponding to its graph label (e.g. the yellow arrow), so as to categorize the graphs by gathering those graphs with the same labels together and isolating those graphs with different labels (e.g. the blue and green arrow) away in the graph updating process. Meanwhile, node representations (e.g. $h'$) will be closer to the mean vector of their potential corresponding Gaussian functions through node updating process.

### 4.3    INFERENCE OF NODES

We infer the categories of the nodes in the graph based on the distance between the node representation $\vec{h}'_i$ obtained from GML and the cluster centers, i.e. mean vector $\vec{\mu}$ of Gaussian functions. We use $C_{node}$ to denote the expected number of node clusters, as in traditional clustering methods this is a hyperparameter that needs to be set by the user. When $C_{node} < C_{graph}$, clustering algorithm such as K-Means (Ding & He, 2002) is used to cluster $C_{graph}$ mean vectors of Gaussian functions to get $C_{node}$ cluster centers. Then, the category $\hat{z}_i$ of the $i$-th node is determined as follows:

$$\hat{z}_i = \arg\min_c distance(\vec{h}'_i, \vec{\mu}_c). \qquad (8)$$

When $C_{node} > C_{graph}$, which is relatively rare in practice, clustering algorithm is used to cluster node representations $\vec{h}'$ to $C_{node}$ clusters directly.

## 5    EXPERIMENTS

In this section, to validate the effectiveness of GMGCN under different cases, we first conduct the experiments on several synthetic datasets. Then we constructed two real-world datasets, the number of graph categories and node categories in these two real-world datasets are both 2.

### 5.1    BASELINES & EXPERIMENTAL SETTINGS

Since WSGC is a new challenge without any baseline work yet, we compare the proposed method with three types of most similar methods: **Feature Clustering** methods without structure information (Jain, 2010; McLachlan & Basford, 1988; Hinton & Salakhutdinov, 2006), **Graph Clustering** methods (Kipf & Welling, 2016; Bo et al., 2020) and **MIL** (Ray & Craven, 2005; Andrews et al., 2002b) methods. We also examine the variant of the proposed method as **ATTGCN**: GMGCN without GML, we replace node inference with multi-head attention mechanisms in the attentive pooling layer, then distinguish the categories of the nodes by comparing attention scores.

We set $\beta = 0.1$ in the hierGAT, the experimental results for $\beta$ equals other values are shown in Appendix C.3.2. The parameters are updated using stochastic gradient descent via Adam algorithm (Kingma & Ba, 2014). The training process is iterated upon 800 epochs for GMGCN. We apply Normalized Mutual Information (NMI) (Strehl & Ghosh, 2002) and Adjusted Rand Index (ARI) (Hubert & Arabie, 1985) to evaluate the graph clustering results on Synthetic Datasets and PHEME Datasets, apply the intersection-over-union (IoU) (Jaccard, 1912), as our evaluation metric on S3DIS Datasets. We run all methods 10 times to avoid extremes and report the average value with a standard deviation. We only report the best results of each type of method in the sequel due to the page limitations. More detailed baseline descriptions and results are available in Appendix C.

### 5.2    GMGCN ON SYNTHETIC DATA

**Synthetic Data Statistics.** We generate several synthetic datasets to examine the proposed WSGC problem with various cases. We name each dataset by the number of graph categories and the

Table 1: Comparison of different types of methods on GC2NC2 (mean(std)).

| methods | single graph | | | | | |
|---|---|---|---|---|---|---|
| | orignal nodes | | new nodes | | new graphs | |
| | NMI | ARI | NMI | ARI | NMI | ARI |
| Feature Clustering | 46.10(0.51) | 42.19(0.79) | 32.07(1.05) | 24.42(0.88) | 45.58(1.41) | 42.86(1.94) |
| Graph Clustering | 80.65(8.18) | 85.51(7.96) | 85.85(8.47) | 87.72(8.52) | 78.72(10.47) | 84.25(10.33) |
| MIL | 76.74(0.00) | 81.67(0.00) | 74.38(0.00) | 79.64(0.00) | 76.61(0.00) | 81.53(0.00) |
| ATTGCN | 11.08(3.36) | 8.53(4.58) | 16.60(2.22) | 10.08(3.30) | 12.11(3.42) | 11.09(5.05) |
| GMGCN | **94.16(0.36)** | **96.81(0.20)** | **96.66(0.42)** | **97.56(0.33)** | **93.88(0.60)** | **96.73(0.29)** |

| methods | hierarchical graph | | | | | |
|---|---|---|---|---|---|---|
| | orignal nodes | | new nodes | | new graphs | |
| | NMI | ARI | NMI | ARI | NMI | ARI |
| ATTGCN | 11.94(2.77) | 8.71(3.34) | 17.18(2.55) | 10.14(2.92) | 12.62(2.92) | 11.67(3.05) |
| GMGCN | **95.68(0.23)** | **97.65(0.12)** | **99.01(0.31)** | **99.28(0.23)** | **95.69(0.34)** | **97.70(0.18)** |

Table 2: Comparison of different types of methods on Multi-classes Synthetic Datasets (mean(std)).

| methods | single graph | | | | | | | |
|---|---|---|---|---|---|---|---|---|
| | GC3NC3 | | GC4NC4 | | GC4NC2-1 | | GC4NC2-2 | |
| | NMI | ARI | NMI | ARI | NMI | ARI | NMI | ARI |
| Feature Clustering | 34.77(0.50) | 32.13(0.58) | 51.10(0.53) | 35.17(0.83) | 34.51(0.95) | 33.21(0.89) | 28.69(0.58) | 26.37(0.68) |
| Graph Clustering | 14.13(3.28) | 10.81(4.42) | 37.11(7.01) | 35.12(7.79) | 3.88(1.50) | 1.05(2.26) | 4.77(2.43) | 4.41(3.18) |
| MIL | 46.92(0.00) | 45.53(0.00) | **61.28(0.00)** | 47.22(0.00) | - | - | - | - |
| ATTGCN | 4.06(3.18) | 3.62(3.61) | 35.67(9.26) | 31.48(8.88) | 2.98(2.12) | 3.30(2.33) | 2.02(0.92) | 2.05(0.93) |
| GMGCN | **51.34(0.77)** | **50.07(1.34)** | 60.01(5.86) | **53.72(9.90)** | **59.55(2.42)** | **66.88(3.21)** | **45.50(3.72)** | **46.18(5.22)** |

| methods | hierarchical graph | | | | | | | |
|---|---|---|---|---|---|---|---|---|
| | GC3NC3 | | GC4NC4 | | GC4NC2-1 | | GC4NC2-2 | |
| | NMI | ARI | NMI | ARI | NMI | ARI | NMI | ARI |
| ATTGCN | 9.78(4.60) | 7.97(3.52) | 36.16(8.71) | 34.44(8.19) | 0.40(0.05) | 0.34(0.04) | 1.72(1.57) | 0.94(0.82) |
| GMGCN | **49.81(1.23)** | **47.10(1.95)** | **62.29(4.94)** | **57.19(7.97)** | **59.67(3.12)** | **67.18(5.24)** | **46.86(4.69)** | **48.00(6.66)** |

number of nodes, for example, we use GC4NC4 to denote a synthetic dataset with the number of graph categories as 4 and the number of node categories as 4. More detailed data statistics and data generation process of these synthetic datasets are detailed in Appendix C.3.1.

**Case 1: Transductive learning.** All graphs are available in the training. Tables 1 and 2 show that GMGCN outperforms all the baselines. In particular, it improves NMI and ARI by an average of more than 13% and 15%, respectively, compared to the best performing baseline. This indicates a significant improvement of the graph clustering guided by graph labels. Since MIL cannot handle an unequal number of graph categories and node categories, it has no results on GC4NC2 datasets.

**Case 2 & Case 3: Inductive learning.** Portions of graphs or nodes are available in the training. For the testing of new nodes and new graphs, we sample 20% of the nodes from each graph and 20% of the graphs in GC2NC2 as testing, respectively. GMGCN also outperforms the other baselines as shown in Table 1. Therefore, GMGCN works for both transductive and inductive settings.

**Case 4: With hierarchical graph.** We regard the inclusion of a hierarchical graph structure as a special case (Case 4) of WSGC. As shown in Table 1 and Table 2, the hierarchical graph structure promotes both the performance of ATTGCN and GMGCN. This indicates that the structure information between graphs is also beneficial to graph clustering.

**Attention vs. GML.** We compare the ATTGCN with GMGCN to verify the importance of GML. The results show that GML greatly promotes GMGCN to identify nodes, while the attention mechanism fails. In addition, a more detailed ablation study is employed to illustrate the importance of each component of GMGCN, due to page limitation we present the experimental results in C.3.3.

**Visualization.** We also visualize the node representations of 4 graphs randomly sampled from GC4NC4 in a 2D space by t-SNE (Van Der Maaten, 2014). As shown in Fig. 2(a)-2(d), GMGCN clearly distinguishes these 4 node categories but the others fail. This indicates that the node representations learned by GMGCN are reasonable for WSGC.

Consequently, the out-performance of GMGCN for all cases imply that the proposed method is a perfect solution for WSGC.

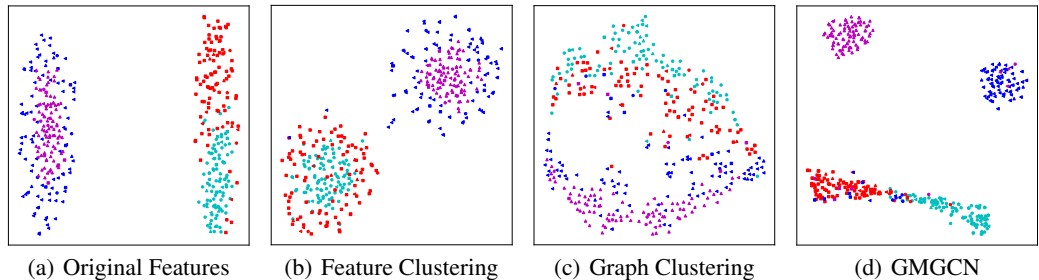

| (a) Original Features | (b) Feature Clustering | (c) Graph Clustering | (d) GMGCN |

Figure 2: 2D visualization of the original node features, the node representations learned by feature clustering, graph clustering, and GMGCN from 4 randomly sampled graphs in GC4NC4. The different shapes represent different graphs and the different colors represent different node categories.

## 5.3 GMGCN ON S3DIS DATA

**Data Description.** S3DIS (Armeni et al., 2016) is composed of 6 large-scale indoor areas, mainly conference rooms, offices and open spaces. Each point is labeled with one of the semantic labels from 13 categories (ceiling, floor, wall, etc. plus clutter). Based on the S3DIS dataset, we construct 13 binary segmentation datasets according to different categories. For each category, the points in the category are annotated with 1, and the rest points are annotated with 0. We treat each point as a node in a graph and apply $k$-NN to construct the edges between points. Each point is represented by a 9-dim vector of each point's 3D spatial coordinates and auxiliary features such as color and normalized location. Each graph is annotated with the label that over 50% of the nodes in the graph belong to. Because there is no potential correlation between each graph, this is a case without a hierarchical graph (Case 4). A detailed data description is presented in Appendix C.4.1.

Table 3: Comparison of different types of methods on S3DIS datasets in terms of IoU (%).

| | training data | | | | | | | | | | | | | |
|---|---|---|---|---|---|---|---|---|---|---|---|---|---|---|
| Methods | Ceiling | Floor | Wall | Beam | Column | Window | Door | Table | Chair | Sofa | Bookcase | Board | Clutter | mIoU |
| Feature Clustering | 41.71 | 41.40 | 32.15 | 31.55 | 32.93 | 33.22 | 31.70 | 36.34 | 32.27 | 46.99 | 31.60 | 30.87 | 31.64 | 34.95 |
| Graph Clustering | 7.11 | 7.87 | 33.46 | 51.13 | 34.29 | 35.05 | 26.38 | 49.82 | 46.70 | **55.60** | 38.42 | 37.97 | 28.77 | 34.81 |
| MIL | 1.75 | 0.12 | 0.19 | 0.00 | 0.00 | 0.00 | 0.00 | 0.00 | 0.00 | 0.00 | 0.69 | 0.00 | 0.40 | 0.24 |
| GMGCN | **74.04** | **87.11** | **42.64** | **53.15** | **42.98** | **47.10** | **50.34** | **50.18** | **53.91** | 52.83 | **46.44** | **41.00** | **45.44** | **52.86** |

| | test data | | | | | | | | | | | | | |
|---|---|---|---|---|---|---|---|---|---|---|---|---|---|---|
| Methods | Ceiling | Floor | Wall | Beam | Column | Window | Door | Table | Chair | Sofa | Bookcase | Board | Clutter | mIoU |
| Feature Clustering | 39.95 | 35.64 | 32.21 | 27.62 | 30.50 | 32.03 | 30.53 | 28.69 | 27.71 | 31.59 | 31.59 | 29.69 | 29.79 | 31.35 |
| Graph Clustering | 64.57 | 13.99 | 36.48 | 20.39 | 34.39 | 19.74 | 37.22 | 23.95 | 18.19 | **44.10** | 36.06 | 31.70 | 26.46 | 31.33 |
| MIL | 0.03 | 0.02 | 0.02 | 0.00 | 0.00 | 0.00 | 0.00 | 0.00 | 0.00 | 0.00 | 0.00 | 0.00 | 0.05 | 0.01 |
| GMGCN | **65.61** | **82.31** | **44.84** | **32.95** | **42.62** | **42.84** | **42.24** | **33.83** | **42.61** | 37.19 | **37.95** | **38.42** | **31.65** | **44.24** |

**Segmentation Results.** We compare the results of each method on the training data and the test data of the S3DIS dataset. As shown in Table 3, the graph clustering methods and GMGCN outperform feature clustering methods and MIL, which demonstrates that structural information plays an essential role in graph clustering. Furthermore, GMGCN outperforms all the baselines in 12 out of 13 datasets, which explicitly reveals the effectiveness of using graph labels to enhance the training of the graph clustering models. In addition, GMGCN outperforms the best-performing baselines by 18% on the training data and by 13% on the test data in terms of mIoU, which further implies a strong inductive learning ability of GMGCN. However, GMGCN does not perform well on the *Sofa* dataset because the dataset containing only 4 graphs in the training is too small for the proposed method to learn enough information from these graphs with the help of graph labels.

## 5.4 GMGCN ON PHEME DATA

**Data Description.** PHEME consists of 5 real-life tweets sets, where each set is related to a piece of breaking news (Zubiaga et al., 2016). Every breaking news includes a lot of rumor and non-rumor topics. A detailed data description is presented in Appendix C.5.1. Based on this dataset, we assume each topic as a graph, where links between users are formed according to their follows/retweets. The link between two graphs is constructed if they have common users. We label the users appearing in more than $M$ rumor topics as abnormal users and the rest as normal users. When $M = 2, 3, 4$, the proposed method all achieves the best results, the detailed results are shown in Appendix C.5.2. In this section, we only show the results when $M = 4$.

Table 4: Comparison of different types of methods on PHEME datasets (mean(std)).

|  | Charlie Hebdo | Ferguson | Germanwings Crash | Ottawa Shooting | Sydney Siege |
|---|---|---|---|---|---|
| Method | NMI | NMI | NMI | NMI | NMI |
| Feature Clustering | 25.46(0.00) | 24.56(0.00) | 44.59(0.00) | 34.56(0.00) | 19.45(0.00) |
| Graph Clustering | 3.26(2.26) | 0.57(0.29) | 0.66(0.50) | 4.49(3.07) | 4.93(1.92) |
| MIL | 5.69(0.00) | 4.08(0.00) | 0.61(0.00) | 0.60(0.00) | 19.68(0.00) |
| Feature Clustering+feat aug | 25.42(0.00) | 24.57(0.00) | 44.84(0.00) | 34.33(0.00) | 19.21(0.00) |
| Graph Clustering+feat aug | 6.08(3.06) | 3.99(3.84) | 4.34(2.25) | 6.22(4.15) | 6.83(4.28) |
| MIL+feat aug | 22.32(0.00) | 0.14(0.00) | 0.28(0.00) | 0.11(0.00) | 19.98(0.00) |
| GMGCN w/o hier | 47.51(3.27) | 48.35(4.08) | 48.85(2.14) | 32.58(3.63) | 41.00(3.93) |
| GMGCN w hier | **53.26(1.62)** | **60.78(0.66)** | **56.63(1.18)** | **41.21(1.15)** | **44.37(0.98)** |
| Method | ARI | ARI | ARI | ARI | ARI |
| Feature Clustering | 23.46(0.00) | 23.23(0.00) | 42.45(0.00) | 32.08(0.00) | 16.45(0.00) |
| Graph Clustering | 6.56(6.30) | 3.02(2.08) | 3.71(1.93) | 10.66(7.68) | 11.78(5.57) |
| MIL | 9.38(0.00) | 13.93(0.00) | 1.10(0.00) | 2.09(0.00) | 39.51(0.00) |
| Feature Clustering+feat aug | 23.41(0.00) | 23.23(0.00) | 48.29(0.00) | 31.85(0.00) | 16.20(0.00) |
| Graph Clustering+feat aug | 10.11(6.02) | 10.12(7.56) | 10.12(4.47) | 8.91(7.99) | 12.65(6.99) |
| MIL+feat aug | 42.51(0.00) | 1.08(0.00) | 0.04(0.00) | 0.04(0.00) | 33.71(0.00) |
| GMGCN w/o hier | **52.38(3.15)** | 55.26(3.21) | 54.95(1.02) | 37.51(2.23) | 37.79(2.18) |
| GMGCN w hier | 51.81(1.63) | **60.30(0.66)** | **56.63(1.18)** | **39.06(1.13)** | **41.32(0.98)** |

**Clustering Results.** As shown in Table 4, GMGCN without hierarchical structure (denoted by GMGCN w/o hier) perfectly resolves WSGC with an average $13\%$ improvement on NMI and a $15\%$ improvement on ARI on these five PHEME datasets compared with the best-performing baseline methods. Furthermore, our method with hierarchical structure (denoted by GMGCN w hier) performs best, which indicates that the interconnections among graphs are helpful for inferring the categories of nodes in the graphs. This significant improvement implies that there exist certain applications in the real world that fulfill the statement of WSGC. In contrast, the other methods perform poorly because WSGC is a totally new clustering problem that differs from feature clustering, graph clustering, or MIL. It should be noted that the best performing feature clustering method in the PHEME dataset, DBSCAN(Ester et al., 1996), is deterministic and no variance is caused, so the standard deviation of feature clustering is 0 shown in Table 4.

**Data Validity.** To show that the artificially constructed data is valid, we conduct a comparative experiment on PHEME datasets by augmenting the node features with a vector, each element of which counts how many graphs of a particular category this node belongs to. In Table 4, *Method+feat aug* means the node features augmented with the vector. Although most clustering methods have been improved by augmenting the node features, they are still generally inferior to GMGCN. This is because although we take into account the number of rumor graphs that each node has appeared when labeling the node, these counts do not participate in the supervised learning, so even if we augment the node features with these counts, the traditional methods still cannot identify the node labels. The experimental results demonstrate that our proposed method can deduce the correct results, and it is valuable to propose a GNN-based method to solve such WSGC problem.

## 6 DISCUSSION AND FUTURE WORK

In this work, we explore an interesting and novel problem: *Weakly Supervised Graph Clustering* (WSGC). In one word, can we identify nodes given the labels of graphs? The proposed WSGC problem has different notions and statements from any of the feature clustering, graph clustering, or multiple-instance learning tasks. To address this new issue, we propose GMGCN with the integration of GMM and GNNs to resolve certain cases of WSGC problem. Experimental results on Synthetic datasets and real-world datasets reveal the need for formulating WSGC and the advantage of the proposed GMGCN over other related baselines. Nevertheless, our study is just an initial step towards the weakly supervised graph learning problems while a variety of extensions are still potential. For example, can we infer the labels of edges or sub-graphs instead of nodes? Can we detect anomaly nodes other than conducting graph clustering? We hope our study will open up a new vein of graph learning and encourage more specifications, solutions, and developments for weakly supervised graph learning.

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

# A    ALGORITHM OF THE PROPOSED METHOD

Algorithm 1 shows the pseudo-codes of GMGCN.

---

**Algorithm 1** GMGCN algorithm

---

**Input:** graph instances set $G$; feature matrix of nodes $\mathbf{X}$; number of graph categories $C_{graph}$; hyper-parameter: $C_{node}$

**Output:** $\{\hat{z}_1^{(1)}, \ldots, \hat{z}_{M_n}^{(1)}; \ldots; \hat{z}_1^{(n)}, \ldots, \hat{z}_{M_n}^{(n)}; \ldots; \hat{z}_1^{(N)}, \ldots, \hat{z}_{M_n}^{(N)}\} \in \{0, 1, \ldots, C_{node} - 1\}$

1: **while** GMGCN not converge **do**
2:     $N$ = number of graphs in $G$
3:     $\vec{\mu}_c$ = mean vector of $c$-th Gaussian weight function
4:     **for** $n = 1, \ldots, N$ **do**
5:         $\mathcal{G}^{(n)}$ = graph structure of $n$-th graph in $G$
6:         $y^{(n)}$ = graph label of $n$-th graph in $G$
7:         $M_n$ = number of nodes in $\mathcal{G}^{(n)}$
8:         **for** $i = 1, \ldots, M_n$ **do**
9:             Update node representation $\vec{h}_i \leftarrow GAT(x_i^{(n)})$
10:            Update node representation $\vec{h}_i' \leftarrow \frac{1}{C_{graph}} \sum_{c=1}^{C_{graph}} w_c(\vec{h}_i, \vec{\mu}_c)$
11:        **end for**
12:        Calculate graph representation $h_g^{(n)} \leftarrow \text{Attn}(\mathbf{H}^{(n)})$
13:        Update graph representation $h_g'^{(n)} \leftarrow \text{hierGAT}(\mathbf{H}_g)$
14:        Calculate consensus loss $L \leftarrow f(\vec{\mu}_c, h_g'^{(n)}, y^{(n)})$
15:    **end for**
16:    Update parameters
17: **end while**
18: Cluster $C_{graph}$ mean vectors into $C_{node}$ mean vectors
19: **for** $n = 1, \ldots, N$ **do**
20:    **for** $i = 1, \ldots, M_n$ **do**
21:        Infer the category of each node $\hat{z}_i^{(n)} \leftarrow \arg\min_c distance(\vec{h}_i', \vec{\mu}_c)$
22:    **end for**
23: **end for**
24: **return** $\{\hat{z}_1^{(1)}, \ldots, \hat{z}_{M_n}^{(1)}; \ldots; \hat{z}_1^{(n)}, \ldots, \hat{z}_{M_n}^{(n)}; \ldots; \hat{z}_1^{(N)}, \ldots, \hat{z}_{M_n}^{(N)}\}$

---

# B    DISCUSSION AND POTENTIAL IMPACT

Since the proposed Weakly Supervised Graph Clustering (WSGC) is a new weakly supervised learning task, we used two applications as examples, but in fact this task is also widely existed in drug discovery, 3D vision and natural language processing, etc. Some potential examples are described below.

In drug discovery, thinking of molecules as graphs in which atoms as nodes (the functional groups can also be treated as super-nodes) and chemical bonds as edges (Gilmer et al., 2017). By applying our method, we can identify the roles of certain nodes, sub-graphs, or equivalently functional groups, in each molecular given its chemical or physical properties.

In the field of computer vision, weakly supervised semantic segmentation and object detection are very popular research areas. In many practical scenarios such as autonomous driving (Arnold et al., 2019; El Madawi et al., 2019) we need to convert the images information into 3D point clouds, which can be further processed with a graph learning based approach. Therefore our proposed method has a lot of space for exploration in such 3D vision scenarios.

In the field of natural language processing, there is a lot of research nowadays that is combined with graph neural networks (Zhang et al., 2019; Linmei et al., 2019). For example, treating words, sentences or paragraphs as nodes and articles as graphs, if we have information about the sentimentality

of each article, we can apply our weakly supervised approach to identify the sentiment expressed in each words, sentences or paragraphs.

## C  EXPERIMENTAL DETAILS

In this section, we present the detailed experimental results corresponding to the Experiment section. First of all, we introduce the baseline methods mentioned in the Experiment section. Then, we present the detailed results on Synthetic, S3DIS and PHEHE datasets.

### C.1  BASELINES

**Feature Clustering Methods:**

- ***K*-means** (Jain, 2010): A classical clustering method that aims to partition data points into $K$ clusters in which each data point belongs to the cluster with the nearest cluster centroid.
- **GMM** (McLachlan & Basford, 1988): A probabilistic model that assumes all the data points are generated from a mixture of a finite number of Gaussian distributions with unknown parameters.
- **DBSCAN** (Ester et al., 1996): A density-based algorithm for discovering clusters in large spatial databases with noise.
- **AE** (Hinton & Salakhutdinov, 2006): A two-stage deep clustering algorithm. We perform GMM on the representations learned by autoencoder.

**Graph Clustering Methods:**

- **GAE & VGAE** (Kipf & Welling, 2016): A structural deep clustering model that combines GCN with the (variational) autoencoder to learn representations. we perform GMM on the representations learned by graph autoencoder.
- **SDCN** (Bo et al., 2020): A structural deep clustering network model that integrates the structural information into deep clustering.

**Multiple Instance Learning:**

- **SIL** (Ray & Craven, 2005): Single-Instance Learning (SIL) is a MIL approach that assigns each instance the label of its bag, creating a supervised learning problem but mislabeling negative instances in positive bags.
- **miSVM** (Andrews et al., 2002b): A SVM-based MIL method. After instance labels have been initialized, an SVM classifier is trained and used to update the label assignation. These two steps are performed iteratively until the label assignation remains unchanged.

We implement $K$-means, GMM and DBSCAN with scikit-learn[3]; AE, GAE, VGAE, SDCN with Pytorch[4]; SIL, miSVM with a Python implementation created by (Doran & Ray, 2014). To be consistent with related works (Bo et al., 2020), the dimension of AE is set to $d$-500-500-2000-10, where $d$ is the dimension of the input data, and the dimension of GAE and VGAE is set to $d$-256-16. For GMGCN, we adopt 2 independent attention mechanisms for the first GATConv layer and use 1 attention mechanism for the second GATConv layer.

### C.2  METRICS

**NMI.** NMI (Normalized Mutual Information) is a normalization of the Mutual Information score to scale the clustering results between 0 (no mutualinformation) and 1 (perfect correlation). It is calculated as follows:

$$NMI(Y, C) = \frac{2 * I(Y, C)}{H(Y) + H(C)}, \tag{9}$$

where $Y$ denotes the class labels, $C$ denotes the cluster labels, $H(\cdot)$ means entropy calculated by $H(Y) = -\sum_{i=1}^{|Y|} P(i)log(P(i))$, $I(Y, C)$ means mutual information between $Y$ and $C$ is calculated

---

[3]https://scikit-learn.org
[4]https://pytorch.org/

by:

$$I(Y, C) = \sum_{i=1}^{|Y|} \sum_{j=1}^{|C|} P(i,j) log(\frac{P(i,j)}{P(i)P(j)}). \tag{10}$$

**ARI** The Rand Index (RI) computes a similarity measure between two clusterings by considering all pairs of samples and counting pairs that are assigned in the same or different clusters in the predicted and true clusterings. It is calculated as follows:

$$RI = \frac{true\_positive + true\_negative}{true\_positive + false\_positive + false\_negative + true\_negative}. \tag{11}$$

The ARI score using the following scheme: $ARI = (RI - Expected\_RI)/(max(RI) - Expected\_RI)$. The adjusted Rand index is thus ensured to have a value close to 0.0 for random labeling independently of the number of clusters and samples and exactly 1.0 when the clusterings are identical (up to a permutation).

**mIoU** IoU is a commonly used metric in semantic segmentation measures similarity between finite sample sets,and is defined as the size of the intersection divided by the size of the union of the sample sets, i.e., $IoU = true\_positive/(true\_positive + false\_positive + false\_negative)$. mIoU (mean Intersection-over-Union) is a common evaluation metric for semantic image segmentation, which first computes the IoU for each semantic class and then computes the average over classes.

## C.3 GMGCN ON SYNTHETIC DATA

### C.3.1 DATA GENERATION & DATA STATISTICS

First, according to the number of node categories $C_{node}$, we generate $C_{node}$ synthetic user groups. The edge relationships between users in same user groups are generated by the Barabási–Albert graph model (Barabási & Albert, 1999). We use Gaussian distribution to randomly generate user features. The mean and standard deviation of different types of users are randomly sampled from [-5, 5] and [1, 10], respectively. Second, we randomly connect users between user groups to simulate the relationship between different types of users. Third, graph structures composed of different types of users is constructed by sampling users in different user groups. In the case that the number of node categories is equal to the number of graph categories, the label of each graph is determined by the major labels of the users in the graph. In other cases, we annotate the same label to graphs with the same percentage of users in each category. Finally, we construct the connections between two graphs if they have more than 5 common users. The data statistics of each Synthetic Dataset are shown in the Table C1.

Table C1: Statistics of Synthetic Datasets.

|  | GC2NC2 | GC3NC3 | GC4NC4 | GC4NC2-1 | GC4NC2-2 |
|---|---|---|---|---|---|
| #Graph categories | 2 | 3 | 4 | 4 | 4 |
| #Node categories | 2 | 3 | 4 | 2 | 2 |
| #Graphs | 100 | 60 | 80 | 120 | 100 |
| #Nodes per graph | 100 | 100 | 100 | 100 | 200 |
| #Features | 20 | 15 | 15 | 15 | 15 |
| Proportion of nodes | 4:1 | 8:1:1 | 85:5:5:5 | 17:3 & 9:1 | 19:1 & 17:3 |

### C.3.2 DETAILED RESULTS

Table C2 and Table C3 show the detailed experimental results of Table 1 and Table 2 in the Experiment section. According to the clustering effectiveness, *K*-Means is selected as the representative of the feature clustering methods, SDCN is selected as the representative of graph clustering methods and SIL is selected as the representative of the MIL methods to be displayed in the Experiment section.

Table C2: Comparison of different methods on GC2NC2 (mean(std)). The bold numbers represent the best results.

| methods | single graph | | | | | |
| --- | --- | --- | --- | --- | --- | --- |
| | orignal nodes | | new nodes | | new graphs | |
| | NMI | ARI | NMI | ARI | NMI | ARI |
| K-Means | 46.10(0.51) | 42.19(0.79) | 32.07(1.05) | 24.42(0.88) | 45.58(1.41) | 42.86(1.94) |
| GMM | 27.86(1.69) | 21.59(2.17) | 23.01(0.97) | 15.63(1.30) | 33.00(4.64) | 27.66(5.06) |
| DBSCAN | 0.41(0.00) | 0.29(0.00) | 0.00(0.00) | 0.00(0.00) | 0.30(0.00) | 0.56(0.00) |
| AE | 5.11(14.33) | 5.36(15.01) | 5.39(16.13) | 5.29(15.94) | 4.73(14.18) | 4.96(14.92) |
| GAE | 37.69(3.69) | 38.80(4.25) | 40.80(4.33) | 38.01(4.45) | 34.38(4.51) | 35.71(5.74) |
| VGAE | 32.06(3.32) | 28.63(3.88) | 33.90(3.44) | 28.24(3.63) | 28.93(2.85) | 25.18(2.89) |
| SDCN | 80.65(8.18) | 85.51(7.96) | 85.85(8.47) | 87.72(8.52) | 78.72(10.47) | 84.25(10.33) |
| SIL | 76.74(0.00) | 81.67(0.00) | 74.38(0.00) | 79.64(0.00) | 76.61(0.00) | 81.53(0.00) |
| miSVM | 30.66(0.00) | 29.25(0.00) | 20.34(0.00) | 22.16(0.00) | 21.05(0.00) | 24.37(0.00) |
| ATTGCN | 11.08(3.36) | 8.53(4.58) | 16.60(2.22) | 10.08(3.30) | 12.11(3.42) | 11.09(5.05) |
| GMGCN | **94.16(0.36)** | **96.81(0.20)** | **96.66(0.42)** | **97.56(0.33)** | **93.88(0.60)** | **96.73(0.29)** |
| methods | hierarchical graph | | | | | |
| | orignal nodes | | new nodes | | new graphs | |
| | NMI | ARI | NMI | ARI | NMI | ARI |
| ATTGCN | 11.94(2.77) | 8.71(3.34) | 17.18(2.55) | 10.14(2.92) | 12.62(2.92) | 11.67(3.05) |
| GMGCN | **95.68(0.23)** | **97.65(0.12)** | **99.01(0.31)** | **99.28(0.23)** | **95.69(0.34)** | **97.70(0.18)** |

Table C3: Comparison of different methods on Multi-classes Synthetic dataset (mean(std)). The bold numbers represent the best results.

| methods | single graph | | | | | | | |
| --- | --- | --- | --- | --- | --- | --- | --- | --- |
| | GC3NC3 | | GC4NC4 | | GC4NC2-1 | | GC4NC2-2 | |
| | NMI | ARI | NMI | ARI | NMI | ARI | NMI | ARI |
| K-Means | 34.77(0.50) | 32.13(0.58) | 51.10(0.53) | 35.17(0.83) | 34.51(0.95) | 33.21(0.89) | 28.69(0.58) | 26.37(0.68) |
| GMM | 32.44(0.81) | 29.02(0.95) | 46.90(0.86) | 29.44(1.18) | 19.75(1.11) | 16.12(1.50) | 21.41(1.99) | 17.57(2.36) |
| DBSCAN | 3.63(0.00) | 4.60(0.00) | 1.76(0.00) | 3.56(0.00) | 0.00(0.00) | 0.00(0.00) | 5.85(0.00) | 2.24(0.00) |
| AE | 15.80(1.47) | 12.14(2.15) | 44.34(1.21) | 44.74(1.34) | 11.37(0.88) | 8.28(1.16) | 20.79(1.37) | 18.51(1.48) |
| GAE | 11.96(1.18) | 7.80(1.71) | 21.46(0.78) | 12.54(0.89) | 8.23(3.43) | 2.47(3.43) | 14.89(1.62) | 13.19(2.02) |
| VGAE | 12.17(1.03) | 8.70(1.78) | 24.75(1.59) | 16.82(1.97) | 8.85(0.78) | 4.01(0.86) | 15.79(1.27) | 13.76(1.63) |
| SDCN | 14.13(3.28) | 10.81(4.42) | 37.11(7.01) | 35.12(7.79) | 3.88(1.50) | 1.05(2.26) | 4.77(2.43) | 4.41(3.18) |
| SIL | 46.92(0.00) | 45.53(0.00) | **61.28(0.00)** | 47.22(0.00) | - | - | - | - |
| miSVM | 35.03(0.00) | 33.47(0.00) | 45.34(0.00) | 31.24(0.00) | - | - | - | - |
| ATTGCN | 4.06(3.18) | 3.62(3.61) | 35.67(9.26) | 31.48(8.88) | 2.98(2.12) | 3.30(2.33) | 2.02(0.92) | 2.05(0.93) |
| GMGCN | **51.34(0.77)** | **50.07(1.34)** | 60.01(5.86) | **53.72(9.90)** | **59.55(2.42)** | **66.88(3.21)** | **45.50(3.72)** | **46.18(5.22)** |
| methods | hierarchical graph | | | | | | | |
| | GC3NC3 | | GC4NC4 | | GC4NC2-1 | | GC4NC2-2 | |
| | NMI | ARI | NMI | ARI | NMI | ARI | NMI | ARI |
| ATTGCN | 9.78(4.60) | 7.97(3.52) | 36.16(8.71) | 34.44(8.19) | 0.40(0.05) | 0.34(0.04) | 1.72(1.57) | 0.94(0.82) |
| GMGCN | **49.81(1.23)** | **47.10(1.95)** | **62.29(4.94)** | **57.19(7.97)** | **59.67(3.12)** | **67.18(5.24)** | **46.86(4.69)** | **48.00(6.66)** |

Fig. C1 shows the results of GMGCN with hierarchical graph when $\beta$ equals different values. As shown in Fig. C1, when $\beta$ is greater than 0 and less than 0.5, GMGCN is more effective. We choose $\beta$ to be equal to 0.1 in the Experiment section.

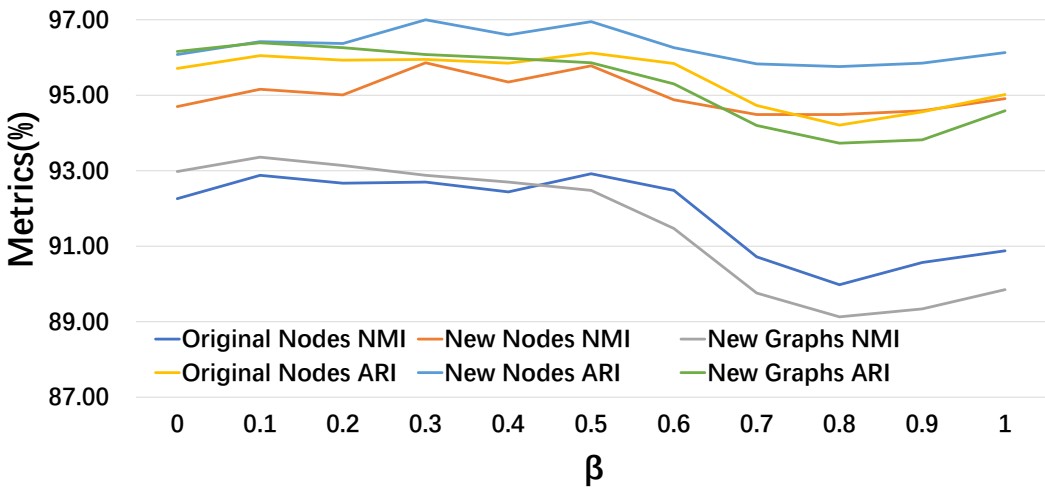

Figure C1: Results of GMGCN with hierarchical graph when $\beta$ equals different values on Synthetic dataset.

Table C4: Ablation Study on GC2NC2 (mean(std)).

|  | w/o 2-layer GAT | w/o GML | w/o Attentive Pooling | w/o hierGAT | w/o consensus loss | GMGCN |
|---|---|---|---|---|---|---|
| NMI | 26.15(2.83) | 11.94(2.77) | 89.22(0.39) | 49.20(23.84) | 50.75(1.19) | 95.68(0.23) |
| ARI | 29.38(2.14) | 8.71(3.34) | 93.89(0.25) | 50.95(29.98) | 54.02(1.83) | 97.65(0.12) |

### C.3.3 ABLATION STUDY.

To verify the importance of each component of GMGCN, we replace each one of the components separately to evaluate its performance and compare it with GMGCN. The results are shown in Table C4, where 'w/o 2-layer GAT' denotes 2-layer GAT is replaced by 2-layer GCN; 'w/o GML' denotes GML-based architecture is replaced by attention-based architecture i.e., ATTGCN; 'w/o Attentive Pooling' denotes attentive pooling is repalced by mean pooling; 'w/o hierGAT' denotes hierGAT is repalced by fully conected layers; 'w/o consensus loss' denotes consensus loss is repalced by cross entropy loss for graph classification.

The best performance of GMGCN indicates that each component of the proposed model plays an important role in predicting the node labels. Specifically, the proposed GML greatly promotes GMGCN to identify nodes, while the attention mechanism fails. When hierGAT is replaced with fully connected layers, the experimental results became very unstable with large standard deviations, illustrating the importance of hierGAT. Consensus loss also greatly improves the model effectiveness because it effectively guides the training of the mean vector of Gaussian functions.

## C.4 GMGCN ON S3DIS DATA

### C.4.1 DATA DESCRIPTION

Distributions of training data and test data in S3DIS datasets are shown in Table C5. In order to maintain the balance of training data, we randomly sample some graphs annotated with 0, and finally make the number of graphs of different categories in the training set equal. Based on this rule, we can see that objects such as *Wall*, *Ceiling* and *Floor* have sufficient graphs for training, whereas objects such as *Sofa* and *Table* have very little graphs for training.

### C.4.2 DETAILED RESULTS

Table C6 shows the detailed experimental results of Table 4 in the Experiment section. Since *Sofa* dataset containing only 4 graphs in the training is too small for the proposed method to learn enough information, GMGCN does not perform well on the dataset. Due to the large number of instances in

Table C5: Data distribution of training set and test set in S3DIS datasets.

| | Ceiling | Floor | Wall | Beam | Column | Window | Door | Table | Chair | Sofa | Bookcase | Board | Clutter |
|---|---|---|---|---|---|---|---|---|---|---|---|---|---|
| | | | | | | Training Data | | | | | | | |
| 1 | 771 | 639 | 2099 | 7 | 115 | 88 | 404 | 7 | 16 | 2 | 325 | 13 | 437 |
| 0 | 771 | 639 | 2099 | 7 | 115 | 88 | 404 | 7 | 16 | 2 | 325 | 13 | 437 |
| total | 1542 | 1278 | 4198 | 14 | 230 | 176 | 808 | 14 | 32 | 4 | 650 | 26 | 874 |
| | | | | | | Test Data | | | | | | | |
| 1 | 414 | 417 | 1270 | 0 | 27 | 93 | 44 | 1 | 0 | 5 | 283 | 0 | 135 |
| 0 | 6342 | 6223 | 1933 | 24 | 282 | 259 | 787 | 2401 | 1783 | 113 | 1369 | 279 | 4231 |
| total | 6756 | 6640 | 3203 | 24 | 309 | 352 | 831 | 2402 | 1783 | 118 | 1652 | 279 | 4366 |

each bag, miSVM requires an extremely large amount of memory for training, so we cannot show its results.

Table C6: Comparison of different types of methods on S3DIS datasets in terms of IoU (%).

| Methods | Ceiling | Floor | Wall | Beam | Column | Window | Door | Table | Chair | Sofa | Bookcase | Board | Clutter | mIoU |
|---|---|---|---|---|---|---|---|---|---|---|---|---|---|---|
| | | | | | | | training data | | | | | | | |
| Kmeans | 41.71 | 41.40 | 32.15 | 31.55 | 32.93 | 33.22 | 31.70 | 36.34 | 32.27 | 46.99 | 31.60 | 30.87 | 31.64 | 34.95 |
| GMM | 43.47 | 42.37 | 33.11 | 26.15 | 31.42 | 31.64 | 33.50 | 18.18 | 26.06 | 32.45 | 32.41 | 30.20 | 32.44 | 31.80 |
| DBSCAN | 30.26 | 31.68 | 27.26 | 33.66 | 29.58 | 28.76 | 28.90 | 32.80 | 30.41 | 31.25 | 29.54 | 31.43 | 31.15 | 30.51 |
| AE | 42.14 | 43.07 | 32.55 | 30.18 | 32.06 | 30.58 | 32.19 | 33.26 | 26.66 | **59.66** | 32.62 | 32.96 | 32.91 | 35.45 |
| GAE | 36.41 | 37.12 | 32.43 | 23.89 | 34.11 | 33.41 | 32.81 | 35.62 | 32.20 | 16.26 | 32.94 | 34.07 | 32.96 | 31.86 |
| VGAE | 37.47 | 36.56 | 32.36 | 32.58 | 33.02 | 36.12 | 31.83 | 34.05 | 23.37 | 32.48 | 33.35 | 31.65 | 32.74 | 32.89 |
| SDCN | 7.11 | 7.87 | 33.46 | 51.13 | 34.29 | 35.05 | 26.38 | 49.82 | 46.70 | 55.60 | 38.42 | 37.97 | 28.77 | 34.81 |
| SIL | 1.75 | 0.12 | 0.19 | 0.00 | 0.00 | 0.00 | 0.00 | 0.00 | 0.00 | 0.00 | 0.69 | 0.00 | 0.40 | 0.24 |
| GMGCN | **74.04** | **87.11** | **42.64** | **53.15** | **42.98** | **47.10** | **50.34** | **50.18** | **53.91** | 52.83 | **46.44** | **41.00** | **45.44** | **52.86** |
| | | | | | | | test data | | | | | | | |
| Kmeans | 39.95 | 35.64 | 32.21 | 27.62 | 30.50 | 32.03 | 30.53 | 28.69 | 27.71 | 31.59 | 31.59 | 29.69 | 29.79 | 31.35 |
| GMM | 40.32 | 36.57 | 33.53 | 26.79 | 33.09 | 32.86 | 31.56 | 28.61 | 28.21 | 30.80 | 31.29 | 29.75 | 29.71 | 31.78 |
| DBSCAN | 36.94 | 37.80 | 28.64 | **46.90** | 37.44 | 32.60 | 39.86 | 43.56 | 44.91 | 40.14 | 35.32 | **41.47** | **42.18** | 39.06 |
| AE | 39.95 | 36.79 | 32.77 | 26.02 | 31.35 | 33.60 | 31.97 | 28.66 | 28.04 | 32.50 | 32.24 | 30.25 | 29.83 | 31.84 |
| GAE | 34.05 | 33.53 | 32.25 | 28.35 | 31.24 | 32.20 | 31.40 | 29.76 | 28.06 | 32.32 | 32.71 | 29.41 | 29.49 | 31.14 |
| VGAE | 34.02 | 34.23 | 32.37 | 29.29 | 30.86 | 33.74 | 30.78 | 29.15 | 27.83 | 30.59 | 32.48 | 30.04 | 29.96 | 31.18 |
| SDCN | 64.57 | 13.99 | 36.48 | 20.39 | 34.39 | 19.74 | 37.22 | 23.95 | 18.19 | **44.10** | 36.06 | 31.70 | 26.46 | 31.33 |
| SIL | 0.03 | 0.02 | 0.02 | 0.00 | 0.00 | 0.00 | 0.00 | 0.00 | 0.00 | 0.00 | 0.00 | 0.00 | 0.05 | 0.01 |
| GMGCN | **65.61** | **82.31** | **44.84** | 32.95 | **42.62** | **42.84** | **42.24** | 33.83 | 42.61 | 37.19 | **37.95** | 38.42 | 31.65 | **44.24** |

## C.5 GMGCN ON PHEME DATA

### C.5.1 DATA DESCRIPTION

PHEME dataset[5] consists of five different news events, all of which attracted substantial attention in the social media and were rife with rumors. The whole dataset is a collection of 5802 tweets, of which 1972 were deemed rumors and 3830 were deemed non-rumors. These tweets are distributed differently across the five events, as shown in Table C7. We also present the distribution of abnormal users and normal users across five events when $M = 2, 3, 4$ in Table C8. The features adopted in PHEME dataset are listed in Table C9.

### C.5.2 DETAILED RESULTS

In PHEME datasets, we label the users appearing in more than $M$ rumor topics as abnormal users, and the rest as normal users. In this part, we present the detailed results on PHEME datasets when $M = 2, 3, 4$. As shown in Table C10 and C11, it is obvious that our proposed method can achieve the best effectiveness on PHEME datasets with different $M$ values. According to the clustering effectiveness, DBSCAN is selected as the representative of the feature clustering methods, GAE is

---

[5]https://figshare.com/articles/PHEME_dataset_of_rumours_and_non-rumours/4010619

selected as the representative of graph clustering methods and SIL is selected as the representative of the MIL methods to be displayed in the Experiment section.

Table C7: Distribution of rumors and non-rumors for the five events in the PHEME dataset.

| Event | Rumors | Non-rumors | Total |
|---|---|---|---|
| Charlie Hebdo | 458(22.0%) | 1621(78.0%) | 2079 |
| Ferguson | 284(24.8%) | 859(75.2%) | 1143 |
| Germanwings Crash | 238(50.7%) | 231(49.3%) | 469 |
| Ottawa Shooting | 470(52.8%) | 420(47.2%) | 890 |
| Sydney Siege | 522(42.8%) | 699(57.2%) | 1221 |
| Total | 1972(34.0%) | 3830(66.0%) | 5802 |

Table C8: Distribution of abnormal users and normal users for the five events in the PHEME dataset when $M = 2, 3, 4$.

| Event | Abnormal users | Normal users | Total |
|---|---|---|---|
| M=2 | | | |
| Charlie Hebdo | 273(13.1%) | 1806(86.9%) | 2079 |
| Ferguson | 256(22.4%) | 887(77.6%) | 1143 |
| Germanwings Crash | 66(14.1%) | 403(85.9%) | 469 |
| Ottawa Shooting | 288(32.4%) | 602(67.6%) | 890 |
| Sydney Siege | 335(27.4%) | 886(72.6%) | 1221 |
| Total | 1218(21.0%) | 4584(79.0%) | 5802 |
| M=3 | | | |
| Charlie Hebdo | 59(2.8%) | 2020(97.2%) | 2079 |
| Ferguson | 50(4.4%) | 1093(95.6%) | 1143 |
| Germanwings Crash | 22 (4.7%) | 447(95.3%) | 469 |
| Ottawa Shooting | 64 (7.2%) | 826(92.8%) | 890 |
| Sydney Siege | 101(8.3%) | 1120(91.7%) | 1221 |
| Total | 296(5.1%) | 5506(94.9%) | 5802 |
| M=4 | | | |
| Charlie Hebdo | 23(1.1%) | 2056(98.9%) | 2079 |
| Ferguson | 13(1.1%) | 1130(98.9%) | 1143 |
| Germanwings Crash | 5(1.1%) | 464(98.9%) | 469 |
| Ottawa Shooting | 15(1.7%) | 875(98.3%) | 890 |
| Sydney Siege | 42(3.4%) | 1179(96.6%) | 1221 |
| Total | 98(1.7%) | 5704(98.3%) | 5802 |

Table C9: Description of features used in PHEME dataset.

| Features | Data Type | Description |
|---|---|---|
| profile_use_background_image | Bool | Whether the user has background image |
| default_profile_image | Bool | Whether the user is using the default profile image |
| verified | Bool | Whether the user's identity is verified by Twitter |
| profile_location | Int | The code of this user's location in the profile |
| followers_count | Int | The number of user's followers |
| listed_count | Int | The number of lists this user jioned |
| statuses_count | Int | The number of tweets posted by this user |
| description | Bool | Whether the user has personal descriptions |
| friends_count | Int | The number of users who have a mutual following relationship with this user |
| favourites_count | Int | The number of user's favourites |
| created_time | Int | The time interval between the user retweeted the relevant tweet and the original tweet was posted |

Table C10: Comparison of different methods on PHEME datasets when $M = 2$ and $M = 3$ (mean(std)). The NMI metric is employed to evaluate each method. The bold numbers represent the best results.

| Method | M=2 | | | | |
|---|---|---|---|---|---|
| | Charlie Hebdo | Ferguson | Germanwings Crash | Ottawa Shooting | Sydney Siege |
| K-Means | 12.90(0.19) | 8.49(0.11) | 25.06(0.57) | 20.16(0.18) | 13.88(0.12) |
| GMM | 10.59(0.26) | 6.79(0.19) | 20.95(0.99) | 16.88(0.55) | 12.25(0.32) |
| DBSCAN | 22.47(0.00) | 20.22(0.00) | **32.95(0.00)** | **27.19(0.00)** | 16.02(0.00) |
| AE | 11.21(0.31) | 12.18(0.66) | 21.76(0.75) | 15.30(0.66) | 11.03(0.44) |
| GAE | 2.20(1.26) | 0.63(0.35) | 1.79(1.81) | 1.84(1.60) | 2.01(0.84) |
| VGAE | 1.79(0.87) | 0.48(0.68) | 1.24(0.69) | 0.99(0.81) | 1.39(0.54) |
| SDCN | 8.69(2.32) | 9.25(2.71) | 9.30(1.90) | 9.31(2.57) | 9.47(0.87) |
| SIL | 8.24(0.00) | 2.73(0.00) | 0.32(0.00) | 3.17(0.00) | 10.49(0.00) |
| miSVM | 11.45(0.00) | 4.71(0.00) | 0.94(0.00) | 2.83(0.00) | 2.12(0.00) |
| GMGCN | **36.69(1.33)** | **25.53(1.66)** | 32.92(1.73) | 24.06(2.35) | **20.74(1.84)** |

| Method | M=3 | | | | |
|---|---|---|---|---|---|
| | Charlie Hebdo | Ferguson | Germanwings Crash | Ottawa Shooting | Sydney Siege |
| K-Means | 10.88(0.12) | 5.63(0.00) | 17.65(0.18) | 18.22(0.30) | 12.80(0.20) |
| GMM | 8.87(0.26) | 2.48(0.35) | 15.05(0.55) | 14.82(0.46) | 11.45(0.60) |
| DBSCAN | 24.54(0.00) | 23.40(0.00) | 39.81(0.00) | **30.88(0.00)** | 17.61(0.00) |
| AE | 9.40(0.32) | 9.12(0.55) | 16.26(0.74) | 15.45(0.58) | 10.97(0.30) |
| GAE | 3.49(2.26) | 0.71(0.68) | 1.65(1.76) | 3.28(3.01) | 1.98(1.38) |
| VGAE | 2.62(0.93) | 0.91(0.70) | 1.27(0.76) | 1.53(1.43) | 1.52(0.82) |
| SDCN | 7.61(4.81) | 8.78(3.43) | 9.51(3.69) | 9.77(3.20) | 9.78(2.42) |
| SIL | 8.24(0.00) | 2.73(0.00) | 0.00(0.00) | 0.48(0.00) | 14.86(0.00) |
| miSVM | 15.07(0.00) | 1.58(0.00) | 0.17(0.00) | 0.79(0.00) | 0.92(0.00) |
| GMGCN | **43.60(2.54)** | **37.07(4.58)** | **40.11(2.72)** | 27.53(1.54) | **37.43(2.30)** |

Table C11: Comparison of different methods on PHEME datasets when $M = 4$ (mean(std)). The bold numbers represent the best results.

| | Charlie Hebdo | Ferguson | Germanwings Crash | Ottawa Shooting | Sydney Siege |
|---|---|---|---|---|---|
| Method | NMI | NMI | NMI | NMI | NMI |
| K-Means | 10.12(0.18) | 4.62(0.10) | 10.38(0.22) | 15.51(0.28) | 11.69(0.16) |
| GMM | 8.10(0.29) | 3.82(0.18) | 8.20(0.49) | 12.27(0.54) | 9.93(0.37) |
| DBSCAN | 25.46(0.00) | 24.56(0.00) | 44.59(0.00) | 34.56(0.00) | 19.45(0.00) |
| AE | 7.15(0.26) | 2.66(0.14) | 6.70(0.44) | 11.36(0.34) | 7.10(0.37) |
| GAE | 3.26(2.26) | 0.57(0.29) | 0.66(0.50) | 4.49(3.07) | 4.93(1.92) |
| VGAE | 2.08(0.46) | 1.08(0.71) | 0.77(0.68) | 4.10(3.93) | 1.38(0.72) |
| SDCN | 1.18(0.63) | 0.74(0.60) | 0.82(0.40) | 1.15(0.60) | 0.06(0.07) |
| SIL | 5.69(0.00) | 4.08(0.00) | 0.61(0.00) | 0.60(0.00) | 19.68(0.00) |
| miSVM | 17.74(0.00) | 0.50(0.00) | 0.00(0.00) | 0.56(0.00) | 0.69(0.00) |
| K-Means+feat aug | 18.48(0.12) | 9.33(0.07) | 12.85(0.32) | 24.82(0.43) | 20.03(0.17) |
| GMM+feat aug | 11.69(0.37) | 7.85(0.21) | 9.64(0.76) | 15.89(0.50) | 13.09(0.43) |
| DBSCAN+feat aug | 25.42(0.00) | 24.57(0.00) | 44.84(0.00) | 34.33(0.00) | 19.21(0.00) |
| AE+feat aug | 10.93(0.26) | 13.20(0.36) | 12.94(0.53) | 9.94(0.42) | 10.05(0.49) |
| GAE+feat aug | 6.08(3.06) | 3.99(3.84) | 4.34(2.25) | 6.22(4.15) | 6.83(4.28) |
| VGAE+feat aug | 5.49(2.29) | 3.45(2.90) | 2.67(2.08) | 5.83(4.09) | 5.45(3.44) |
| SDCN+feat aug | 14.66(2.42) | 14.93(4.43) | 39.14(11.78) | 18.98(2.36) | 11.00(2.07) |
| SIL+feat aug | 22.32(0.00) | 0.14(0.00) | 0.28(0.00) | 0.11(0.00) | 19.98(0.00) |
| miSVM+feat aug | 17.74(0.00) | 0.50(0.00) | 0.00(0.00) | 0.56(0.00) | 0.69(0.00) |
| GMGCN w/o hier | 47.51(3.27) | 48.35(4.08) | 48.85(2.14) | 32.58(3.63) | 41.00(3.93) |
| GMGCN w hier | **53.26(1.62)** | **60.78(0.66)** | **56.63(1.18)** | **41.21(1.15)** | **44.37(0.98)** |
| Method | ARI | ARI | ARI | ARI | ARI |
| K-Means | 8.20(0.20) | 3.33(0.15) | 9.14(0.29) | 13.27(0.30) | 8.75(0.19) |
| GMM | 5.52(0.40) | 2.06(0.24) | 6.21(0.69) | 9.00(0.65) | 6.35(0.48) |
| DBSCAN | 23.46(0.00) | 23.23(0.00) | 42.45(0.00) | 32.08(0.00) | 16.45(0.00) |
| AE | 4.36(0.31) | 0.34(0.24) | 4.08(0.57) | 7.76(0.50) | 2.73(0.45) |
| GAE | 6.56(6.30) | 3.02(2.08) | 3.71(1.93) | 10.66(7.68) | 11.78(5.57) |
| VGAE | 2.30(1.74) | 4.04(2.24) | 3.41(2.57) | 9.56(8.68) | 2.10(3.20) |
| SDCN | 1.03(1.00) | 0.74(0.90) | 0.92(0.66) | 0.76(1.01) | 0.27(0.36) |
| SIL | 9.38(0.00) | 13.93(0.00) | 1.10(0.00) | 2.09(0.00) | 39.51(0.00) |
| miSVM | 32.28(0.00) | 2.80(0.00) | 0.01(0.00) | 0.31(0.00) | 1.88(0.00) |
| K-Means+feat aug | 18.01(0.14) | 9.64(0.09) | 11.96(0.34) | 23.85(0.45) | 18.58(0.21) |
| GM+feat aug | 10.06(0.42) | 7.57(0.29) | 7.83(0.93) | 13.40(0.64) | 10.29(0.50) |
| DBSCAN+feat aug | 23.41(0.00) | 23.23(0.00) | 48.29(0.00) | 31.85(0.00) | 16.20(0.00) |
| AE+feat aug | 7.67(0.28) | 11.42(0.44) | 9.57(0.67) | 4.92(0.59) | 5.02(0.63) |
| GAE+feat aug | 10.11(6.02) | 10.12(7.56) | 10.12(4.47) | 8.91(7.99) | 12.65(6.99) |
| VGAE+feat aug | 6.61(5.27) | 9.15(6.51) | 7.00(5.31) | 11.46(8.15) | 10.50(8.87) |
| SDCN+feat aug | 10.73(2.96) | 10.96(4.38) | 36.47(12.04) | 14.00(2.71) | 4.84(2.47) |
| SIL+feat aug | 42.51(0.00) | 1.08(0.00) | 0.04(0.00) | 0.04(0.00) | 33.71(0.00) |
| miSVM+feat aug | 32.28(0.00) | 2.80(0.00) | 0.01(0.00) | 0.31(0.00) | 1.88(0.00) |
| GMGCN w/o hier | **52.38(3.15)** | 55.26(3.21) | 54.95(1.02) | 37.51(2.23) | 37.79(2.18) |
| GMGCN w hier | 51.81(1.63) | **60.30(0.66)** | **56.63(1.18)** | **39.06(1.13)** | **41.32(0.98)** |

