# OpenReview forum: "Weakly Supervised Graph Clustering"
_ICLR.cc/2022/Conference — ICLR 2022 Submitted_

### Official Review · Reviewer_PUKs · 2021-11-02

**Correctness:** 3
**Technical Novelty And Significance:** 3
**Empirical Novelty And Significance:** 3
**Recommendation:** 5
**Confidence:** 3

**Main Review:**

Strengths:
1. The problem is well-motivated.
2. The proposed method is reasonable.
3. Extensive experiments are given.
Weaknesses:
1. The comparison methods are not convincing. It is better to employ some recent techniques.
2. The technique is not as novel as the authors claim.

**Summary Of The Paper:**

This paper investigates the graph clustering problem with the aid of graph labels. GMM and GCN are used to solve this problem. A new loss is also designed. Extensive experiments are conducted to show the effectiveness of the proposed method.

**Summary Of The Review:**

1. There are some closely related works in the literature. The authors should discuss the contributions more clearly.
2. Does this paper "Semi-Supervised Graph Classification: A Hierarchical Graph Perspective" study the same problem？
3. Is the proposed method related to multi-view/multi-layer graph clustering?
4. There are some writing errors.

---

> ### Author Response · Authors · 2021-11-18
> **Thank you for the constructive feedback**
>
> **To Reviewer PUKs:**
>
> Thank you for your valuable suggestions. We have revised our paper according to your suggestions and marked it in red in the paper. For each of your questions, our response is as follows.
>
> >Q1: The comparison methods are not convincing. It is better to employ some recent techniques.
>
> R1: Since in this paper we aim to solve a new task for weakly supervised learning, but there was no suitable method that could be adopted. We have tried our best to find these comparison methods. If you could provide what methods are more suitable, we would be very appreciative and will compare the proposed method with them in the revised version.
>
> >Q2: The technique is not as novel as the authors claim.
>
> R2: We argue that the proposed method contains innovative techniques: GML and consensus loss for weakly supervised graph clustering modeling. As shown in the experimental results of the ablation study in Section G1.2 in the general responses above. GML and consensus loss indeed significantly promote the proposed method. More importantly, our major contribution lies in the new challenge for weakly supervised learning on the graph that we proposed—WSGC, which, as mentioned in our paper, contributes to many applications, and deserves more attention from the community.
>
> >Q3: There are some closely related works in the literature. The authors should discuss the contributions more clearly.
>
> R3: Thank you for your suggestion. In this paper, we propose a new challenge for weakly supervised learning. As far as we know, two kinds of settings are close to the new challenge which are graph clustering and weakly supervised learning. However, related works in graph clustering are not applicable to the setting of this new task, as they do not use graph labels. On the other hand, we have a comparison with the weakly supervised method MIL, but actually MIL is not applicable to this weakly supervised graph clustering task due to it assumes i.i.d among the instances. If we have any missing literature, we would appreciate it if you could point them out for us.
>
> >Q4: Does this paper "Semi-Supervised Graph Classification: A Hierarchical Graph Perspective" study the same problem？
>
> R4: This paper is to identify the label of the graphs based on the labels of the nodes in the graphs. While we are doing the opposite thing, i.e., identifying the labels of the nodes in the graphs based on the graph labels.
>
> >Q5: Is the proposed method related to multi-view/multi-layer graph clustering?
>
> R5: Multi-view/multi-layer graph clustering is to solve the unsupervised graph clustering task by extracting multi-view/multi-layer through multiple relationships between nodes. While we are solving the proposed weakly supervised graph clustering task by training models with coarse-grained graph labels. These are two different tasks.
>
> >Q6: There are some writing errors.
>
> R6: Thank you for your careful review. We have gone through our paper again and corrected these typos in the revised version.

---

> ### Author Response · Authors · 2021-11-19
> **Welcome to comment**
>
> Dear reviewer,
>
> We have gone through our paper again and corrected the typos in the revised version. Differences from semi-supervised graph classification and multi-view/multi-layer graph clustering are also provided. Please feel free to ask any questions.
>
> Thanks

---

> ### Comment · Reviewer_PUKs · 2021-11-25
> **I  have read the rebuttal**
>
> I updated my score.

---

> > ### Author Response · Authors · 2021-11-25
> > **Thank you for your feedback**
> >
> > We appreciate your effort to help us improve the paper. We noticed that you have downgraded your score. Could you kindly tell us which concern we have not addressed clearly enough so that we can improve it?

---

> ### Author Response · Authors · 2021-11-29
> **Could you please tell us the reason why you downgraded the score?**
>
> Dear reviewer PUKs:
>
> Could you kindly tell us which concern we have not addressed clearly enough so that we can improve it? Since you downgraded our score, we should at least know the weakness of our paper so that we could improve it.
>
> Thanks,
>
> The authors

---

> ### Author Response · Authors · 2021-11-30
> **The end of the discussion phase is approaching**
>
> Dear reviewer PUKs,
>
> Thank you for your comments and suggestions on our paper. We are still willing to know what you think we could further improve our paper although the deadline is coming. Any concrete suggestions would help us to do better. It is so upset to us that you downgraded the score without any reason.
>
> Best regards,
>
> The authors

---

### Official Review · Reviewer_Khg3 · 2021-11-03

**Correctness:** 4
**Technical Novelty And Significance:** 3
**Empirical Novelty And Significance:** 3
**Recommendation:** 6
**Confidence:** 2

**Main Review:**

Strength

1) The paper is well written and structured. Four cases were presented, and comprehensive experimental results on synthetic data are presented for each of the four cases.
2) The twitter example provides a good practical use case of the proposed algorithm.

Weakness

I'd love to have clarity on a few things:
1) Is it assumed that C_node < = C_graph? Do we assume C_node is known? When C_node is unknown, how do we determine it?
2) Is there a real world example where interconnections among graphs exist and are helpful for graph clustering? Given both real-world datasets are single graph case, I can't think of an example that puts case 4 in use.
3) It would be more clear to have an algorithm part, where inputs, outputs, and steps of algorithms are presented.
4) It is not clear to me why we need two datasets of GC4NC2 for experiments and the major difference between them.
5) Is there a minimum number of graphs that the authors recommend for the proposed algorithm? The authors claim that the sofa example didn't do well (relative to other methods) due to small sample size.

Minor points
1) typo: page 3, section 3.1, where C_{node} -1 is the expected number of node categories --> where C_{node} is the expected number of node categories
2) It would be more clear if the authors explain NMI and mIoU in the last paragraph of the introduction section.

**Summary Of The Paper:**

The authors propose a new Gaussian Mixture Graph Convolutional Network approach to perform graph clustering with the assistance of graph labels. Experimental results suggest that the addition of graph labels boosts performance over traditional clustering methods, and over multiple-instance learning.

**Summary Of The Review:**

The problem is novel and the authors have demonstrated that having coarse-grain labels on the whole graph can assist with clustering on a node level. The comments for improvement are mainly on clarifying some details of applying the proposed algorithm.

---

> ### Author Response · Authors · 2021-11-18
> **Thank you for the constructive feedback**
>
> **To Reviewer khg3:**
>
> Thank you for your valuable suggestions. We have revised our paper according to your suggestions and marked it in red in the paper. For each of your questions, our response is as follows.
>
> >Q1: Is it assumed that $C_{node} < = C_{graph}$? Do we assume $C_{node}$ is known? When $C_{node}$ is unknown, how do we determine it?
>
> R1: Actually, in our problem definition, it is not necessary that $C_{node} < = C_{graph}$, when $C_{node} > C_{graph}$, classical cluster methods could be applied to the output of GML layer to get the node inference. We use $C_{node}$ in the paper to denote the number of clusters, as in the traditional clustering method this is a hyperparameter that needs to be set by the user. In practice, we may have some prior knowledge for reference to set this hyperparameter. For example, in the rumor detection setting, Since the number of labels of the graph has two categories, rumor and non-rumor, we divide the users into abnormal users who spread rumors and other normal users, so we set  $C_{node}$  to 2. Generally, we can set it to be equal to or slightly larger than the number of graph categories, and then find the appropriate number of clusters by adjusting this hyperparameter. We have made this clearer in Section 4.3, Inference of Nodes.
>
>
> >Q2: Is there a real world example where interconnections among graphs exist and are helpful for graph clustering? Given both real-world datasets are single graph case, I can't think of an example that puts case 4 in use.
>
> R2: Thanks for the comments. To demonstrate the effectiveness of our method on real-world data with the interconnections among graphs, we construct the hierarchical graph structure based on the PHEME dataset. The connections between two graphs are constructed if they have common users. For detailed experimental results, please refer to Section G1.1, Real-world Hierarchical Graph, in the general responses.
>
>
> >Q3: It would be more clear to have an algorithm part, where inputs, outputs, and steps of algorithms are presented.
>
> R3: Thanks to your valuable suggestion, we have added the algorithm of the proposed method in the Appendix of the revised version due to the page limitation.
>
> >Q4: It is not clear to me why we need two datasets of GC4NC2 for experiments and the major difference between them.
>
> R4: As shown in Table A in the Appendix, in both GC4NC2 datasets, the total number of graphs is different, the total number of nodes in each graph is different, and the proportion of nodes in each category is also different in each graph. Actually, we exhibit 2 different GC4NC2 datasets in order to demonstrate that our model can handle different cases. We have added more detailed statistics in the Appendix of the revised version.
>
> >Q5: Is there a minimum number of graphs that the authors recommend for the proposed algorithm? The authors claim that the sofa example didn't do well (relative to other methods) due to small sample size.
>
> R5: Evaluating the bound on the number of training samples is difficult, which is related to the specific data. As shown in Table 3, GMGCN exhibits greater enhancement compared to baseline methods on datasets with more training samples, such as Ceiling and Floor. Moreover, it also performs well even for Beam and Table, which have only 14 samples. Therefore, without specific experiments, we cannot directly give the minimum recommended number of graphs.
>
> >Q6: typo: page 3, section 3.1, where $C_{node} -1$ is the expected number of node categories --> where $C_{node}$ is the expected number of node categories
>
> R6: Thank you for your careful review. We have corrected it in the revised version.
>
> >Q7: It would be more clear if the authors explain NMI and mIoU in the last paragraph of the introduction section.
>
> R7: Thanks for your suggestion. NMI (Normalized Mutual Information) is a normalization of the Mutual Information score to scale the clustering results between 0 (no mutual information) and 1 (perfect correlation). IoU (intersection-over-union) is a commonly used metric in semantic segmentation and is defined as the size of the intersection divided by the size of the union of the sample sets. mIoU means first computes the IoU for each semantic class and then computes the average over classes. We have explained NMI and mIoU in the Introduction section and added the calculation procedure in the Appendix to make it clearer.

---

> ### Author Response · Authors · 2021-11-19
> **Welcome to comment**
>
> Dear reviewer,
>
> We have conducted the experiments on the real-world dataset with hierarchical graph to demonstrate the effectiveness of the proposed method. An algorithm is also provided in the revised version. Please feel free to ask any questions.
>
> Thanks

---

> ### Author Response · Authors · 2021-11-29
> **The end of the discussion phase is approaching**
>
> Dear reviewer khg3:
>
> We sincerely appreciate your positive comments that the paper is well written and structured, and the Twitter example provides a good practical use case of the proposed algorithm. During the rebuttal period, we have made every effort to faithfully address all your comments. Here, we briefly summarize the major updates as follows:
>
> Regarding the guidance of determining the number of clusters (Q 1), we have clarified it is a hyperparameter that needs to be set by the user and provided some naïve method to set this hyperparameter. We have made this clearer in Section 4.3 Inference of Nodes of the revision.
>
> Regarding the lack of real-world applications with interactions between graphs (Q2), we have constructed the hierarchical graph structure based on the PHEME dataset by connecting the graphs that have common users. The best performance of the proposed approach illustrates the effectiveness of our model in real-world applications with graph interactions. The experimental results have been additionally included in Table 4 of the revision.
>
> Regarding the lack of the algorithm (Q3), we have added the algorithm of the proposed method in the Appendix of the revision.
>
> Regarding the confusion of having 2 GC4NC2 datasets (Q4), we have added more detailed statistics in the Appendix of the revision to illustrate the differences between these two datasets.
>
> Regarding the typo in section3.1 (Q6), we have corrected it in the revision.
>
> Regarding the explanation of the NMI and mIoU in the Introduction (Q7), We have explained NMI and mIoU in the Introduction section and added the calculation procedure in the Appendix to make it clearer.
>
> Could you please go over our full responses and the revision? We really appreciate your insightful and constructive comments.
>
> Thanks,
>
> The authors

---

> ### Comment · Reviewer_Khg3 · 2021-11-30
> **Thank you for addressing my comments**
>
> I wanted to thank the authors for addressing my questions and providing additional experiments. The additional experiments confirmed that using hierarchical graphs did boost the clustering performance. I would like to keep my original score, since I see the contribution of using graph labels to help inform node-level clustering. For further improvements, I think the authors can give more examples/use cases for the proposed algorithm, especially on where the graph labels come from. I think twitter is a good example, since the topic comes labelled, while on S3DIS data, graph is labelled with majority of the node groups, which makes it impossible to obtain the graph labels since we don't know the node labels in practical examples.

---

### Official Review · Reviewer_inpd · 2021-11-07

**Correctness:** 3
**Technical Novelty And Significance:** 3
**Empirical Novelty And Significance:** 3
**Recommendation:** 6
**Confidence:** 2

**Main Review:**

Pros:

1. The paper is well organized, and generally easy to read.
2. The idea of incorporating coarse-grained graph label to assist graph clustering is somewhat interesting, and may be useful in some specific scenarios.
3. Experiments also demonstrating the superiority of the proposed WSGC task over the traditional GC task.


Cons:
1. The current title of the submission seems overclaimed. As weakly supervised graph clustering may also include some other settings such as partial supervision, distant supervision, etc, it seems overclaimed to name the paper title as weakly supervised graph clustering. To me, it will be much better to highlight incorporating graph-level side information, which is the key contribution of this paper.
2. Although the graph labels have been shown to bring significant performance gains in their experiments, it is not clear to me what kind of graph labels are effective for graph clustering? Are they task specific? Do we need to manually design such graph labels?
3. In Table 3, it is interesting that for most categories the GMGCN approach achieves the best performance, but for the sofa category, its performance is much lower than the baselines. Could you please give some analysis to explain the reasons of the performance difference?
4. In Figure 1, what does node updating denote?


**Summary Of The Paper:**

This paper studies a new problem named weakly supervised graph clustering (WSGC), where the goal is to cluster nodes better with some graph-level side information, i.e., graph labels.

Most previous studies for graph clustering primarily consider node/edge information and their connections, but ignore graph-level side information, which might be easily accessible in real scenarios. Inspired by this observation, this paper proposed to incorporate graph-level labels into graph clustering, and formulate the new problem as weakly supervised graph clustering. Based on the new problem, the authors further proposed a simple yet effective framework based on Gaussian mixture model (GMM) and graph convolutional network (GCN) named Gaussian Mixture Graph Convolutional Network (GMGCN) for learning node representations. Moreover, they also introduced a consensus loss for graph labels and employed Gaussian Mixture  Layer to infer the category of each node.

Experiments on both synthetic and real-world datasets demonstrate the effectiveness of the proposed GMGCN approach.


**Summary Of The Review:**

Overall, I think the idea of incorporating coarse-grained graph label to assist graph clustering is intuitive and somewhat interesting, and the proposed model also generally makes sense to me. But I still have some concerns regarding the title, the choice of graph labels, and the performance analysis.

---

> ### Author Response · Authors · 2021-11-18
> **Thank you for the constructive feedback**
>
> **To Reviewer inpd:**
>
> Thank you for your valuable suggestions. We have revised our paper according to your suggestions and marked it in red in the paper. For each of your questions, our response is as follows.
>
> >Q1: The current title of the submission seems overclaimed. As weakly supervised graph clustering may also include some other settings such as partial supervision, distant supervision, etc, it seems overclaimed to name the paper title as weakly supervised graph clustering. To me, it will be much better to highlight incorporating graph-level side information, which is the key contribution of this paper.
>
> R1: Thanks for your suggestion, we have changed the title to a more accurate one i.e., Graph Clustering with the Weakly Supervision from Graph Labels.
>
> >Q2: It is not clear to me what kind of graph labels are effective for graph clustering? Are they task-specific? Do we need to manually design such graph labels?
>
> R2: The graph labels do not need to be designed manually and it is task-specific. The proposed method aims to infer node categories based on graph labels that are semantically related to potential node labels. For example, a social media network with a graph label of rumor or non-rumor will semantically indicate that the network contains normal users and abnormal users who propagate rumors.
>
> >Q3: In Table3, for the sofa category, its performance is much lower than the baselines. Could you please give some analysis to explain the reasons of the performance difference?
>
> R3: We conjecture that GMGCN does not perform well on the Sofa dataset because the dataset containing only 4 graphs in the training is too small for the proposed method to learn enough information from these graphs with the help of graph labels. Obviously, on ceilings, floors, walls, and doors with more training data, our proposed GMGCN method obtains greater improvement compared to the baseline methods. The data distribution of S3DIS datasets is available in Table C5 in the Appendix.
>
> >Q4: In Figure 1, what does node updating denote?
>
> R4: Sorry for the confusion. In Figure 1, what we want to express is that when consensus loss guides the model training through backward propagation, there is a parameter updating process that affects the node-level modeling, and we refer to it as node updating.

---

> > ### Comment · Reviewer_inpd · 2021-11-25
> > **Thanks for your efforts**
> >
> > I think the authors have addressed most of my concerns, and the current version looks OK to me. I still think the idea of incorporating coarse-grained graph label to assist graph clustering is interesting, and thus would like to keep my original score.

---

> > > ### Author Response · Authors · 2021-11-25
> > > **Thank you very much for your feedback**
> > >
> > > Dear reviewer,
> > >
> > > Thank you for your valuable comments. Would you mind providing more suggestions for the further improvement of our work?
> > >
> > > Best regards,
> > >
> > > The authors

---

> ### Author Response · Authors · 2021-11-19
> **Welcome to comment**
>
> Dear reviewer,
>
> We have revised the title to highlight the inclusion of graph-level side information. A more detailed node inference process is also provided in the revised version. Please feel free to ask any questions.
>
> Thanks

---

### Official Review · Reviewer_r2ym · 2021-11-08

**Correctness:** 3
**Technical Novelty And Significance:** 2
**Empirical Novelty And Significance:** 2
**Recommendation:** 5
**Confidence:** 3

**Main Review:**

Strengths
- The proposed methods outperform baseline models on the given datasets and synthetic datasets.
- Two levels of graphs (one for graphs of multiple categories and the other for the hypergraph) are used and those levels are connected through Gaussian Mixture embeddings.

Weaknesses
- Despite the complicated process and a lot of notations, definitions, relationships, and explanations for notations are not sufficient or there exist some typos so that the manuscript is not easy to follow.
- Not knowing the number of clusters is a common problem, but the guidance to tackle that problem is not suggested.
- Despite the long and complicated process, the contribution or the necessity of each step is not clear. Ablation study would be helpful.
- In particular, the contribution of weak supervision is not clear. Without the clear evidence, a weakly supervised method would not be verified over an unsupervised method.

Details
- Eq (4) is not well-defined. First of all, the number of clusters should be C_node, not C_graph. Second, the output of w_c seems to be the vector, while Eq (4) does not necessarily indicate the full Gaussian model. Would \Theta_GLM indicate the inverse of covariance matrix so that Eq. (4) means the entriy-wise Guassian model?
- Authors describe that Eq. (5) does not have the training parameters, but \alpha^{heir} itself contains many weight coefficients. The description should be clearer to clarify the distinction between training parameters and the others.
- Consensus loss is based on the linear sum of the distance to cluster center and the distance to graph center. Then, what is the role of weak supervision if there is no use of true labels?
- GLM-based clustering methods needs to be also compared as baseline methods.

**Summary Of The Paper:**

Authors provide the framework that combines Graph Attention Networks (GAT) and Gaussian Mixture Models (GMM) to tackle the weakly supervised graph clustering model. In particular, the consensus loss, a new loss function designed by the authors, is used for the entire framework. Authors also address different kinds of weak supervision settings and show the performance of the proposed framework for each case on synthetic datasets.

**Summary Of The Review:**

While each step of the entire process is fair, the necessity of each step is not sufficiently verified. Despite the hypergraph setting and the weak supervision setting, authors show the experiments on multi-view graphs or partitioned graphs (for real-world datasets), and do not show the benefits of weak supervision. Furthermore, the proposed method should be compared with Guassian-embedding based models.

---

> ### Author Response · Authors · 2021-11-18
> **Thank you for the constructive feedback**
>
> **To Reviewer r2ym:**
>
> Thank you for your valuable suggestions. We have revised our paper according to your suggestions and marked it in red in the paper. For each of your questions, our response is as follows.
>
> >Q1: Despite the complicated process and a lot of notations, definitions, relationships, and explanations for notations are not sufficient or there exist some typos so that the manuscript is not easy to follow.
>
> R1: Thank you for your suggestion. We have added more explanations in Section 3.1 Notations and Section 4.3 Inference of Nodes. We explained the meaning of $d$ in Section 3.1 Notation, and the difference between $C_{graph}$ and $C_{node}$ in Section 4.3 Inference of Nodes. In addition, we have corrected typos to make it clearer in the revised version. If there is still something we have not made clear, we would appreciate it if you could point them out for us.
>
> >Q2: Not knowing the number of clusters is a common problem, but the guidance to tackle that problem is not suggested.
>
> R2: Yes, same as traditional clustering methods, $C_{node}$ in our paper, which denotes the number of clusters, is a hyperparameter that needs to be set by the user. In practice, we may have some prior knowledge as a reference to set this hyperparameter.
> For example, in the rumor detection setting, since the number of labels of the graph has two categories, rumor and non-rumor, we divide the users into abnormal users who spread rumors and other normal users, so we set $C_{node}$ to 2 during the inference. Generally, the number of node categories is selected to be equal to or slightly larger than the number of graph categories. Thank you for your suggestion. We have made this clearer in Section 4.3, Inference of Nodes.
>
> >Q3: Despite the long and complicated process, the contribution or the necessity of each step is not clear. An ablation study would be helpful.
>
> R3: Thank you for your valuable suggestions. We have illustrated the importance of each component of the proposed method with an ablation study. The detailed experimental settings and results are shown in Section G1.2, Ablation Study, in the general responses. The best performance of GMGCN indicates that each component of the proposed model plays an important role in predicting node labels. Specifically, the proposed GML greatly promotes GMGCN to identify nodes, while the substituted attention mechanism fails. When hierGAT is replaced with fully connected layers, the experimental results became very unstable with large standard deviations, illustrating the importance of hierGAT. In addition, the proposed consensus loss also greatly improves the model effectiveness because it effectively guides the training of the mean vector of Gaussian functions.
>
> >Q4: In particular, the contribution of weak supervision is not clear. Without the clear evidence, a weakly supervised method would not be verified over an unsupervised method.
>
>
> R4: It is because there exists such a situation in practice that whether we can cluster the nodes better with some graph-level side information. Hence, we define a new problem to satisfy this setting, named as a weakly supervised graph clustering problem. While without graph labels, this problem naturally degenerates to an unsupervised clustering problem. So we compare with unsupervised clustering methods to indicate the improvement of the proposed method on the basis of unsupervised methods with the assistant of graph-level side information.
>
> >Q5: Eq (4) is not well-defined. First of all, the number of clusters should be $C_{node}$, not $C_{graph}$. Second, the output of $w_c$ seems to be the vector, while Eq (4) does not necessarily indicate the full Gaussian model. Would $\Theta_{GML}$ indicate the inverse of covariance matrix so that Eq. (4) means the entry-wise Guassian model?
>
> R5: Sorry for the misleading here.
>
> 1. In fact, it is $C_{graph}$ in Eq.(4), and as described in Section 4.3, $C_{graph}$ mean vectors are clustered into $C_{node}$ mean vectors in the process of node inference.
>
> 2. Our intention is to have $\Theta_{GML}$ denote the learning parameters of the Gaussian function, which should be expressed as $w_c(h_i; \Theta_{GML})$ in the paper, we have corrected it to $w_c(h_i)$ in the revised version for clarity, thank you for your careful review.
>
> >Q6: Authors describe that Eq. (5) does not have the training parameters, but $\alpha^{heir}$ itself contains many weight coefficients. The description should be clearer to clarify the distinction between training parameters and the others.
>
> R6: Thank you for your careful review. We intended to clarify that there are no more parameters leading to spatial transformations (like $\mathbf{\Theta}$ in Eq. (1)). We have clarified this in the revised version.

---

> > ### Author Response · Authors · 2021-11-18
> > **Thank you for the constructive feedback**
> >
> > >Q7: Consensus loss is based on the linear sum of the distance to cluster center and the distance to graph center. Then, what is the role of weak supervision if there is no use of true labels?
> >
> > R7: Given your statement that there is no real label, we are not sure if it means graph labels or node labels. Therefore, we discuss both two possible cases here.
> >
> > 1. If without the true graph labels. The task degenerates to feature clustering or graph clustering and is out of our problem settings. In addition, consensus loss is intended to make the graph representation of each graph closer to the mean vector of the Gaussian function corresponding to its graph label. As mentioned in Section 4.2 in the paper, $\ell_{con}:=cross\\_entropy(\mathbf{S}, \mathbf{y})$. Hence, the consensus loss cannot be calculated without the true graph labels.
> >
> > 2. If without the true node labels. We indeed did not use node labels. The role of weakly supervised learning is to assist in inferring node labels by training the model under the supervision of graph labels.
> >
> > If we have misunderstood your question, please point it out to us.
> >
> > >Q8: The proposed method should be compared with Guassian-embedding based models.
> >
> > R8: Sorry, we do not know what Gaussian-embedding-based models refer to. We would appreciate it if you could tell us about these models and we will compare the proposed method with them in the revised version.

---

> ### Author Response · Authors · 2021-11-19
> **Welcome to comment**
>
> Dear reviewer,
>
> We have elaborated on how to determine the number of clusters in a revised paper. In addition, we have conducted an ablation study to illustrate the contribution of each component of the proposed method. Please feel free to ask any questions.
>
> Thanks

---

> ### Author Response · Authors · 2021-11-29
> **The end of the discussion phase is approaching**
>
> Dear reviewer r2ym:
>
> We sincerely appreciate your comments. During the rebuttal period, we have made every effort to faithfully address all your comments. Here, we briefly summarize the major updates as follows:
>
> Regarding the definitions and explanations for notations (Q1), we have added more explanations in Section 3.1 Notations and Section 4.3 Inference of Nodes of the revision.
>
> Regarding the guidance of determining the number of clusters (Q 2), we have clarified it is a hyperparameter that needs to be set by the user and provided some naïve method to set this hyperparameter. We have made this clearer in Section 4.3 Inference of Nodes of the revision.
>
> Regarding the contribution of each step of the model (Q3), we have illustrated the importance of each component of the proposed method with an ablation study. The detailed experimental settings and results have been added in the Appendix C.3.3 of the revision.
>
> Regarding the not well-defined equation (Q5) and wrong-described parameters (Q6), we have corrected these errors in the revision.
>
> Could you please go over our full responses and the revision? We really appreciate your insightful and constructive comments.
>
> Thanks,
>
> The authors

---

### Official Review · Reviewer_oDis · 2021-11-09

**Correctness:** 2
**Technical Novelty And Significance:** 2
**Empirical Novelty And Significance:** 2
**Recommendation:** 5
**Confidence:** 4

**Main Review:**

Even though this manuscript reads well and tries to define a new problem and a solution for the proposed problem, I have some major concerns described below.

According to the definition of the proposed problem, one should have node features, node labels, interactions between the nodes, the graph labels, and possibly the higher-level interactions between the graphs. I wonder if one can grasp all this rich information in practical applications. More importantly, it is not clear how the graph label affects the predictions of node labels. In experiments, the authors consider only one type of real-world data from Twitter, which does not include the interactions between graphs. Without the interactions between graphs (i.e., the hierarchical graph), we can treat each graph as an independent graph and apply well-known graph clustering or node classification methods such as GCN, GraphSAGE or GIN.

For the examples or applications of the proposed problem, the authors mention detecting frauds or abnormal users. Since there is substantial literature for anomaly detection, the proposed method should have more general applications other than anomaly detection. Unfortunately, it is hard for me to think of realistic scenarios or interesting applications of the proposed method.

Overall, the proposed method seems to make a simple problem more complicated. To make the proposed method convincing, more general practical applications should be shown where all the inputs the problem require are provided. Also, the authors should show whether each of the components plays a critical role in predicting the node labels and analyze how each component affects the results.

* Minor comments: In Section 3.1, $d$ is not defined.

**Summary Of The Paper:**

This manuscript proposes a problem defined as weakly supervised graph clustering. Given a set of graphs and node features, the problem is to predict node labels in each of the graphs. Some variations can be made depending on selecting the targets whose labels are supposed to be predicted.

**Summary Of The Review:**

I reject this paper because (1) it is hard to find practical applications of the proposed method; one can solve the problem more easily in many practical applications, (2) it is not clear how each of the inputs affects the predictions, (3) the experiments do not include diverse real-world datasets.

---

> ### Author Response · Authors · 2021-11-18
> **Thank you for the constructive feedback**
>
> **To Reviewer oDis:**
>
> We thank the reviewer for the provided comments. After reading the comments seriously, we are afraid that the reviewer might misunderstand the problem definition of our paper.  Based on the misunderstanding, the proposed Q1, Q4, Q5 and Q6 might be inapplicable for our paper. We will try our best to eliminate the misunderstanding via the following responses, and sincerely hope that the reviewer raises any further questions if he/she is still confused by our responses.
>
> >Q1: According to the definition of the proposed problem, one should have node features, node labels, interactions between the nodes, the graph labels and possibly the higher-level interactions between the graphs. I wonder if one can grasp all this rich information in practical applications?
>
> R1: There might be a misunderstanding here that our model **does not require node labels** during the training. The proposed model only needs node features, interactions between the nodes, the graph labels and possibly the higher-level interactions between the graphs for training in a weakly supervised manner without seeing any information about the node labels. In addition, we regard the inclusion of a hierarchical graph structure as a special case (Case 4) rather than as an indispensable component of the proposed model. We present our model in different cases and hope that future researchers would choose the appropriate one according to the specific scenario. Therefore, instead of requiring the full information, the proposed method can be tailored and requires less information regarding practical applications. For example, in social media, we only need the graph labels of the social relationship graphs to infer the category of each user, instead of training a node classification model with the user labels.
>
>
> >Q2: It is not clear how the graph label affects the predictions of node labels.
>
>
> R2: It might not be clear enough to explain how the node inference works in Section 4.3.
> When the graph label guides the model training through backward propagation, the parameter updating process will affect the node-level modeling, resulting in node representations affected by the graph labels being closer to the mean vectors of their potential corresponding Gaussian functions. In the process of node inference, we infer the category of each node based on the distance between the node representation and mean vectors. Therefore, we can infer the categories of the nodes in a weakly supervised manner without node labels. We have elaborated this process more clearly in the Model Training Guidance section of the revised version.
>
>
> >Q3: The authors consider only one type of realworld data which does not include the interactions between graphs.
>
>
> R3: Thanks for the comments. As stated in R1, we regard the inclusion of a hierarchical graph structure as a special case (Case 4) rather than as an indispensable component of the proposed model. Nevertheless, it is interesting to demonstrate the effectiveness of our method on real-world data with the interactions between graphs. Hence, we construct the hierarchical graph structure based on the PHEME dataset by connecting the graphs that have common users. For experimental results, please refer to Section G1.1, the Real-world Hierarchical Graph, in the general responses.
>
>
> >Q4: Without the interactions between graphs, we can treat each graph as an independent graph and apply well-nown graph clustering or node classification methods such as GCN, GraphSAGE or GIN.
>
> R4: There might be the same misunderstanding as Q1 here. Our problem setting is different from either graph clustering or node classification. First, we have compared the graph clustering methods with the proposed method, but the graph clustering methods do not perform better than GMGCN due to the lack of graph-level information as supervision for model training. Secondly, node classification methods, such as GCN, GraphSAGE, or GIN, are not capable of the problem if we only know the graph labels but not the node labels during the training. Therefore, we firstly introduce the weakly supervised graph learning problem and propose a novel approach: GMGCN which is trained for predicting node-level labels only with the assistant of graph-level labels.

---

> > ### Author Response · Authors · 2021-11-18
> > **Thank you for the constructive feedback**
> >
> > >Q5: The proposed method should have more general applications other than anomaly detection. It is hard to think of realistic scenarios or interesting applications of the proposed method.
> >
> > R5: Thank you for your suggestions. We have shown another application as point cloud segmentation in the Experiments section in addition to the social network anomaly user detection. We agree that the potential application of this new weakly supervised learning problem needs to be thoroughly elaborated. In this vein, we provide several potential examples below:
> >
> >
> > Some potential examples are described below.
> > 1. In drug discovery, thinking of molecules as graphs in which atoms as nodes (the functional groups can also be treated as super-nodes) and chemical bonds as edges [1]. By applying our method, we can identify the roles of certain nodes, sub-graphs, or equivalently functional groups, in each molecular given its chemical or physical properties.
> >
> > 2. In the field of computer vision, weakly supervised semantic segmentation and object detection are very popular research areas. In many practical scenarios such as autonomous driving [2][3] we need to convert the images information into 3D point clouds, which can be further processed with a graph learning-based approach. Therefore, our proposed method has a lot of space for exploration in such 3D vision scenarios.
> >
> > 3. In the field of natural language processing, there is a lot of research nowadays that is combined with graph neural networks [4][5]. For example, treating words, sentences, or paragraphs as nodes and articles as graphs, if we have information about the sentimentality of each article, we can apply our weakly supervised approach to identify the sentiment expressed in each word, sentence, or paragraph.
> >
> > We have added these examples in the Discussion and Potential Impact section in the Appendix.
> >
> > [1] Gilmer, Justin, et al. "Neural message passing for quantum chemistry." International conference on machine learning. PMLR, 2017.
> >
> > [2] El Madawi, Khaled, et al. "Rgb and lidar fusion based 3d semantic segmentation for autonomous driving." 2019 IEEE Intelligent Transportation Systems Conference (ITSC). IEEE, 2019.
> >
> > [3] Arnold, Eduardo, et al. "A survey on 3d object detection methods for autonomous driving applications." IEEE Transactions on Intelligent Transportation Systems 20.10 (2019): 3782-3795.
> >
> > [4] Zhang, Chen, Qiuchi Li, and Dawei Song. "Aspect-based sentiment classification with aspect-specific graph convolutional networks." arXiv preprint arXiv:1909.03477 (2019).
> >
> > [5] Linmei, Hu, et al. "Heterogeneous graph attention networks for semi-supervised short text classification." Proceedings of the 2019 Conference on Empirical Methods in Natural Language Processing and the 9th International Joint Conference on Natural Language Processing (EMNLP-IJCNLP). 2019.
> >
> > >Q6: The proposed method seems to make a simple problem more complicated.
> >
> > R6: We would like to emphasize that the problem that the proposed method is trying to resolve is **NOT** a simple problem since the problem needs a model to identify the category of each node when only graph-level supervision information is available. Therefore, this problem is not solvable by simple node classification methods because no node labels are available. Furthermore, if we simply formulate the problem as traditional unsupervised clustering methods or weakly supervised methods, they do not perform well on this problem as shown in our experiments. We therefore propose this novel weakly supervised graph clustering task, as a new task that has not received attention in the field of either weakly supervised learning or graph learning but is still interesting and important.
> >
> > >Q7: The authors should show whether each of the components plays a critical role in predicting the node labels and analyze how each component affects the results.
> >
> > R7: Thank you for your valuable suggestions. We have illustrated the importance of each component of the proposed method with an ablation study. The detailed experimental settings and results are shown in Section G1.2, the Ablation Study in the general responses. The best performance of GMGCN indicates that each component of the proposed model plays an important role in predicting node labels. Specifically, the proposed GML greatly promotes GMGCN to identify nodes. When hierGAT is replaced with fully connected layers, the experimental results became very unstable with large standard deviations, illustrating the importance of hierGAT. In addition, the proposed consensus loss also greatly improves the model effectiveness because it effectively guides the training of the mean vector of Gaussian functions.
> >
> >
> > >Q8: Minor comments: In Section 3.1, $d$ is not defined.
> >
> > R8: Thank you for your careful review. $d$ denotes the number of dimensions of node original features in Section 3.1. We have clarified it in the revised version.

---

> ### Author Response · Authors · 2021-11-19
> **Welcome to comment**
>
> Dear reviewer,
>
> We have conducted the experiments on the real-world dataset with hierarchical graph to demonstrate the effectiveness of our method. Besides, an ablation study is conducted to illustrate how each component affects the node inference results. Please feel free to ask any questions.
>
> Thanks.

---

> ### Author Response · Authors · 2021-11-29
> **The end of the discussion phase is approaching**
>
> Dear reviewer oDis:
>
> We sincerely appreciate your comments. During the rebuttal period, we have made every effort to faithfully address all your comments. Here, we briefly summarize the major updates as follows:
>
> Regarding the problem definition (Q1, Q4, Q6), we have clarified that our model **does not require node labels** during the training in the responses. The elimination of this misunderstanding also demonstrates that we are indeed presenting a new problem that needs a model to identify the category of each node when only graph-level supervision information is available rather than making a simple problem more complicated.
>
> Regarding how the graph label affects the predictions of node labels (Q 2), we have further clarified the model training guidance in Section 4.2 of the revision.
>
> Regarding the lack of real-world applications with interactions between graphs (Q3), we have constructed the hierarchical graph structure based on the PHEME dataset by connecting the graphs that have common users. The best performance of the proposed approach illustrates the effectiveness of our model in real-world applications with graph interactions. The experimental results have been additionally included in Table 4 of the revision.
>
> Regarding more general applications (Q5), we have added more realistic applications in the Appendix of the revision.
>
> Regarding the contribution of each part of the model (Q7), we have illustrated the importance of each component of the proposed method with an ablation study. The detailed experimental settings and results have been added in the Appendix C.3.3 of the revision.
>
> Could you please go over our full responses and the revision? We really appreciate your insightful and constructive comments.
>
> Thanks,
>
> The authors

---

> > ### Comment · Reviewer_oDis · 2021-11-30
> > **Thanks for the responses!**
> >
> > Since the authors' responses clarified some issues, I slightly increased my score.

---

### Author Response · Authors · 2021-11-18
**General Responses: Additional Experiments**

**G1 General Responses: Additional Experiments**

We sincerely thank all reviewers for their valuable comments. Below, as several reviewers request an experiment on real-world hierarchical graph datasets and the ablation study for the proposed method, we first propose the additional experiments here. Then we provide point-by-point responses to each reviewer separately. Notably, we have revised the paper to address all reviewers’ suggestions, since revisions are allowed and encouraged during the rebuttal period according to the ICLR official guidance.

**G1.1 Real-world Hierarchical Graph**

The reviewers oDis and khg3 asked about the efficiency of our method on the real-world dataset with the hierarchical graph. To demonstrate the effectiveness of our method on real-world data with the interactions between graphs. We construct the hierarchical graph structure based on the PHEME dataset. The connections between two graphs are constructed if they have common users. We let 'baseline' denote the result of the best-performed baseline; 'w/o hier' denote the structure without interactions between graphs and 'w hier' denote the structure with interactions between graphs.

Our method with interactions between graphs performs best and gains a huge improvement over the method without interactions between graphs. It indicates that the interactions between graphs are helpful for inferring the categories of nodes in this social media application. We have added the results of this real-world hierarchical data to the revised version.

|   NMI      | charliehebdo |    ferguson   | germanwings-crash | ottawashooting | sydneysiege |
|----|----|----|----|----|----|
| baseline |  25.46(0.00)   | 24.56(0.00) |       44.59(0.00)        |    34.56(0.00)    |  19.68(0.00)  |
| w/o hier  | 47.51(3.27)   |  48.35(4.08) |       48.85(2.14)        |    32.58(3.63)    |  41.00(3.93)  |
| w hier     | **53.26(1.62)**    | **60.78(0.66)** |       **56.63(1.18)**        |    **41.21(1.15)**    |  **44.37(0.98)**  |

|   ARI       | charliehebdo |   ferguson   | germanwings-crash | ottawashooting | sydneysiege |
|----|----|----|----|----|----|
| baseline |  23.46(0.00)  | 23.23(0.00) |       42.45(0.00)        |   32.08(0.00)     |  39.51(0.00)  |
| w/o hier  |  **52.38(3.15)**  | 55.26(3.21) |       54.95(1.02)        |   37.51(2.23)     |  37.79(2.18)  |
| w hier     |  51.81(1.63) | **60.30(0.66)** |       **55.24(1.45)**        |   **39.06(1.13)**    |   **41.32(0.98)**  |

**G1.2 Ablation Study**

The reviewers oDis and r2ym asked the importance of each component of the proposed method. To illustrate how each component affects the results, we conduct an ablation study. Since GMGCN consists of five components: 2-layer GAT, GML, Attentive Pooling, hierGAT and consensus loss, we replace each one of the components separately to evaluate its performance and compare it with GMGCN. We conduct experiments on the GC2NC2 dataset, NMI and ARI are used to evaluate the performance of each component. The denotations and results are shown below.

1. w/o 2-layer GAT:  2-layer GAT is replaced by  2-layer GCN.

2. w/o GML: GML-based architecture is replaced by attention-based architecture i.e. ATTGCN mentioned in the paper.

3. w/o Attentive Pooling: Attentive pooling is replaced by mean pooling.

4. w/o hierGAT: hierGAT is replaced by fully connected layers.

5. w/o consensus loss: consensus loss is replaced by cross-entropy for graph classification.

|         | w/o 2-layer GAT |   w/o GML |  w/o Attentive Pooling  | w/o hierGAT |  w/o consensus loss |  GMGCN  |
|----|----|----|----|----|----|----|
| NMI |    26.15(2.83)       |11.94(2.77)|          89.22(0.39)         |49.20(23.84) |         50.75(1.19)       |95.68(0.23)|
| ARI |     29.38(2.14)       | 8.71(3.34) |           93.89(0.25)        |50.95(29.98)|         54.02(1.83)       |97.65(0.12)|


The best performance of GMGCN indicates that each component of the proposed model plays an important role in predicting node labels. Specifically, the proposed GML greatly promotes GMGCN to identify nodes, while the attention mechanism does not. When hierGAT is replaced with fully connected layers, the experimental results became very unstable with large standard deviations, illustrating the importance of hierGAT. The proposed consensus loss also greatly improves the model effectiveness because it effectively guides the training of the mean vectors of Gaussian functions. We have added these experimental results and analyses in the Appendix.

---

### Author Response · Authors · 2021-11-23
**Welcome to comment**

Dear Reviewers,

Thanks very much for your valuable comments to help us to improve our work! We are wondering whether all of your concerns have been addressed properly. We would be glad to answer any further questions you may have after reviewing the feedback.

Best regards,

The authors.

---

### Author Response · Authors · 2021-11-29
**The end of the discussion phase is approaching**

Dear Reviewers,

Could you please go over our responses and the revision? We have responded to your comments and faithfully reflected them in the revision, and provided additional experimental results that you have requested. We sincerely thank you for your time and efforts in reviewing our paper, and your insightful and constructive comments.

BEST REGARDS,

The authors

---

### Note · Authors · 2024-06-05
**Submission Withdrawn by the Authors**

I have read and agree with the venue's withdrawal policy on behalf of myself and my co-authors.

---

### Decision · Program_Chairs · 2022-01-20

**Decision:**

Reject

**Comment:**

Existing methods for graph clustering usually use node/edge information, but ignore graph-level information. This paper proposes incorporating graph-level labels into graph clustering and formulating the new problem as weakly supervised graph clustering.  The paper further proposes Gaussian Mixture Graph Convolutional Network (GMGCN) framework for the task.  Experimental results on several datasets demonstrate the effectiveness of the method.

The authors are very active in answering questions by the reviewers.  They have successfully addressed some of the issues. However, there are still questions that remain unaddressed. The submission is not of the quality of ICLR papers.

Strength
* A new method is proposed.
* The proposed methods outperform baseline models on the given datasets and synthetic datasets.

Weakness
* The explanations are not clear enough.  Although the authors provide detailed responses to the reviews,  the problems indicated by the reviewers are still not well addressed.
* The proposed method seems to be too complicated.
* It is not clear why the proposed method works.
* The problem studied might not be realistic.

----
Here is a summary of the reviewers' final comments.

* Reviewer oDis slightly increased the score。

* Reviewer r2ym says“I read responses to my concerns and others, but except for some clarifying statements and notations, authors' responses are not convincing enough. Also, while I now understand the concept of proposed work better than before, I do not think that it is explained and presented well enough.”

* Reviewer inpd says “would like to keep my original score”.